# High-entropy engineering of the crystal and electronic structures in a Dirac material

Antu Laha[1,6], Suguru Yoshida [1,2,6] ✉, Francisco Marques dos Santos Vieira[3], Hemian Yi[1], Seng Huat Lee [1,2], Sai Venkata Gayathri Ayyagari [3], Yingdong Guan[1], Lujin Min [1,3], Jose Gonzalez Jimenez[4], Leixin Miao[3], David Graf [5], Saugata Sarker [3], Weiwei Xie [4], Nasim Alem [3], Venkatraman Gopalan [3], Cui-Zu Chang [1], Ismaila Dabo [1,3] ✉ & Zhiqiang Mao [1,2,3] ✉

Dirac and Weyl semimetals are a central topic of contemporary condensed matter physics, and the discovery of new compounds with Dirac/Weyl electronic states is crucial to the advancement of topological materials and quantum technologies. Here we show a widely applicable strategy that uses high configuration entropy to engineer relativistic electronic states. We take the $A$MnSb$_2$ ($A$ = Ba, Sr, Ca, Eu, and Yb) Dirac material family as an example and demonstrate that mixing of Ba, Sr, Ca, Eu and Yb at the $A$ site generates the compound (Ba$_{0.38}$Sr$_{0.14}$Ca$_{0.16}$Eu$_{0.16}$Yb$_{0.16}$)MnSb$_2$ (denoted as $A^5$MnSb$_2$), giving access to a polar structure with a space group that is not present in any of the parent compounds. $A^5$MnSb$_2$ is an entropy-stabilized phase that preserves its linear band dispersion despite considerable lattice disorder. Although both $A^5$MnSb$_2$ and $A$MnSb$_2$ have quasi-two-dimensional crystal structures, the two-dimensional Dirac states in the pristine $A$MnSb$_2$ evolve into a highly anisotropic quasi-three-dimensional Dirac state triggered by local structure distortions in the high-entropy phase, which is revealed by Shubnikov–de Haas oscillations measurements.

Topological materials like graphene and three-dimensional Dirac and Weyl semimetals host electrons with linear energy-momentum dispersion near the Fermi energy[1,2]. This sort of electronic state creates a number of intriguing physical properties, including high transport mobility[3,4], large magnetoresistance[5], topological quantum transport such as chiral anomaly[6], among others. The discovery of a new topological material often attracts immense research interest since it may exhibit an unusual topological phenomenon[7–11]. An exotic topological state can also appear in existing materials when the crystal symmetry deviates from equilibrium. For example, the three-dimensional Dirac semimetal (3D DSM), Cd$_3$As$_2$[12], exhibits a change from tetragonal to monoclinic symmetry under high pressure, which drives it into a topological superconducting state otherwise inaccessible[13–15]. Given that 3D DSMs are in the vicinity of topological phase transition[13], they offer rich opportunities for reaching hidden topological states via perturbations like lattice distortions.

The magnitude of lattice distortion also plays a pivotal role in controlling materials' properties, as exemplified by the spin-valley locked state[16] observed in BaMn$X_2$ ($X$ = Sb and Bi)[17–19]. Both compounds consist of the $X_4$ layers forming $X$–$X$ zig-zag chains due to the $X$-atom displacement and, as a result, adopt a polar space group $I2mm$. In contrast to the qualitative similarity in real space, their valley

[1]Department of Physics, Pennsylvania State University, University Park, PA, USA. [2]2D Crystal Consortium, Materials Research Institute, Pennsylvania State University, University Park, PA, USA. [3]Department of Materials Science and Engineering, Pennsylvania State University, University Park, PA, USA. [4]Department of Chemistry, Michigan State University, East Lansing, MI, USA. [5]National High Magnetic Field Laboratory, Tallahassee, FL, USA. [6]These authors contributed equally: Antu Laha, Suguru Yoshida. ✉e-mail: suguru.yoshida0224@gmail.com; dabo@psu.edu; zim1@psu.edu

electronic states are distinct. The spin valley degeneracy is of two for BaMnSb$_2$[17,18], but four for BaMnBi$_2$[19]. Such a variation of spin valley degeneracy originates from the quantitative difference in the magnitude of structural distortions; the lattice distortion of BaMnSb$_2$ is ten times larger than that of BaMnBi$_2$[19]. Therefore, structural distortions not only drive a unique phase transition, but their amplitude can also provide a knob to tune the electronic state of 3D DSMs.

How can we manipulate distortion in a bulk system? Solid-solution approach provides a possible means to do so. However, an accessible range of lattice distortion by this approach just follows Vegard's law[20] and rarely goes beyond distortions of the end members. In addition, the emergence of another polymorph, which may host a disparate topological state, can hardly be expected in solid solutions especially when the end members are isosymmetric.

Here, we introduce a high-entropy approach[21,22] to DSMs as a way to overcome the limitation of the solid-solution approach and demonstrate that the high-entropy approach can yield both new crystal and electronic structures with focusing on $A$MnSb$_2$ ($A$ = Ba, Sr, Ca, Eu, and Yb)[23–28] as a prototypical example. $A$MnSb$_2$ does not adopt a unique structure but shows three distinct structural polymorphs depending on the $A$-cation size and lattice distortion. We have grown homogeneous single crystals of a high-entropy phase, (Ba$_{0.38}$Sr$_{0.14}$Ca$_{0.16}$Eu$_{0.16}$Yb$_{0.16}$)MnSb$_2$ (denoted as $A^5$MnSb$_2$). Our experimental and group-theoretical structural analyses reveal that the high-entropy phase adopts a new crystal structure with space group $P2_1mn$, which is not present in the related parent compounds. Even though high-entropy materials have recently been extensively studied across materials science[29–36], there exist limited examples where the high-entropy approach results in a phase absent in the parent compounds[37]. Further, our work shows that $A^5$MnSb$_2$ not only preserves the Dirac dispersion but also exhibits carrier mobility as high as those in the parent compounds despite the random mixing of $A$ cations. Remarkably, we also find that the severe lattice distortions present in the high-entropy phase lead the Dirac state in $A^5$MnSb$_2$ to be of quasi-3D character, in sharp contrast to the quasi-2D Dirac state in $A$MnSb$_2$. These results demonstrate that the high-entropy approach provides an additional means to discover unforeseen relativistic electronic states in topological matter.

## Results
### Crystal structure of the high-entropy phase

The $A$MnSb$_2$ ($A$ = Ba, Sr, Ca, Eu, and Yb) series exhibits several different types of structures; the identity of the $A$ cation dictates the stacking of MnSb$_4$ layers[38]. For smaller (Ca, Yb) and slightly larger (Sr, Eu) cations, the adjacent MnSb$_4$ layers are directly aligned[26,39,40], and the undistorted aristotype structure has $P4/nmm$ symmetry [Fig. 1a]. In contrast, a larger cation (Ba) drives anti-alignment between MnSb$_4$ blocks in adjacent layers (or one can regard the layers as being offset from each other by $\frac{1}{2}$[110]), which leads to $I4/mmm$ symmetry with doubled unit cell along the $c$ direction [Fig. 1b]. Because of lattice distortion mentioned below, none of the compounds in $A$MnSb$_2$ series adopt the $I4/mmm$ structure, but some of the Bi-based analogs, i.e., SrMnBi$_2$ and EuMnBi$_2$, crystallize in this tetragonal symmetry[41,42]. $A$MnSb$_2$ further undergoes orthorhombic distortions except for YbMnSb$_2$, where Sb exhibits displacements relative to the square lattice, thus resulting in the formation of Sb–Sb zig-zag chains [Fig. 1c]. Despite the shifts of Sb atoms, the Sb displacements are anti-parallel with the same magnitude in the structures with aligned MnSb$_4$ layers [Fig. 1a] due to its symmetry, giving rise to the nonpolar $Pcmn$ phase [Fig. 1c]. In the anti-aligned structures [Fig. 1b], however, no symmetry restriction is imposed on the anti-parallel Sb displacements to be equal in magnitude. As a result, the local electric dipoles accompanied by the Sb displacements no longer cancel out, which produces net polar symmetry, $I2mm$ [Fig. 1c], as identified in BaMnSb$_2$[17,18].

$A$MnSb$_2$ series offers a rich diversity in structures and ionic radii of the $A$ cations. We first checked whether any pairs of the five elements

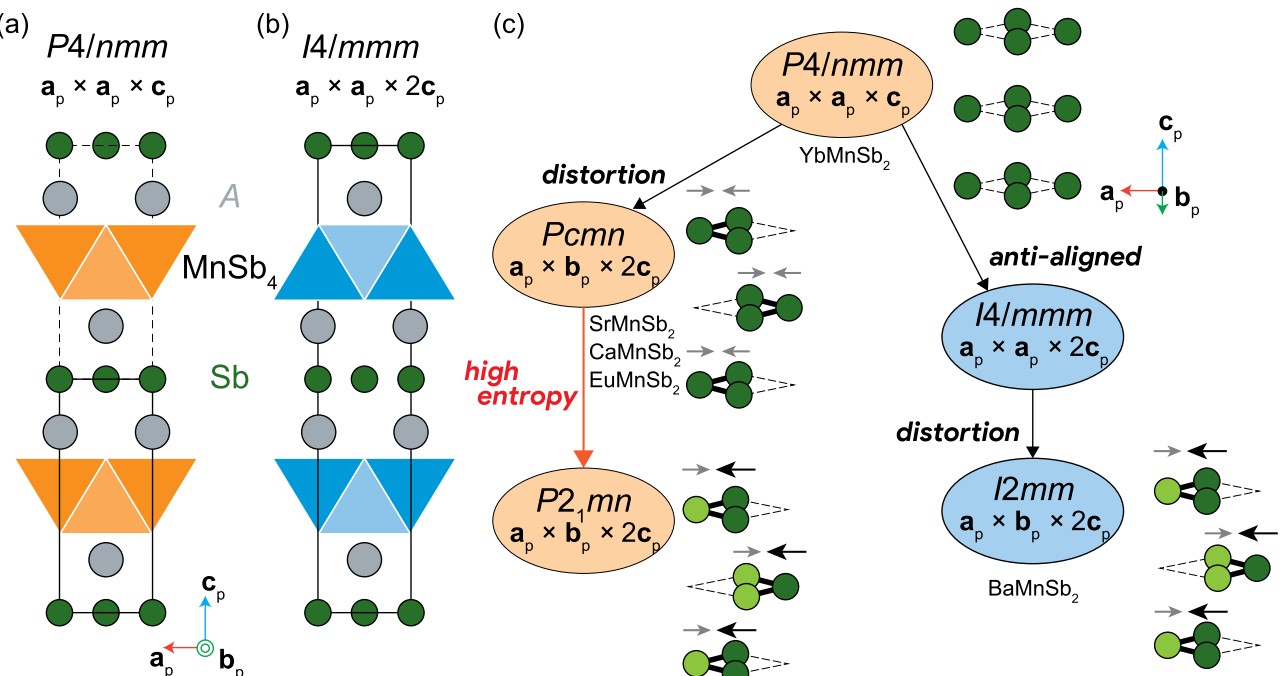

**Fig. 1 | Schematics of the crystal structures of $A$MnSb$_2$ ($A$ = Ba, Sr, Ca, Eu, and Yb) and $A^5$MnSb$_2$.** **a** $P4/nmm$ and **b** $I4/mmm$ aristotype structures of $A$MnSb$_2$, where $\mathbf{a}_p$, $\mathbf{b}_p$, and $\mathbf{c}_p$ represent the lattice vectors of the $P4/nmm$ phase. Black solid lines indicate the unit cells. **c** Treelike diagram showing space groups assigned for the polymorphs of the $A$MnSb$_2$ series so far and for the new structure of $A^5$MnSb$_2$. Sb$_4$ layers are depicted, where dashed lines highlight the square shape. The Sb$_4$ layers of $Pcmn$, $I2mm$, and $P2_1mn$ structures involve Sb displacements (illustrated by arrows, whose size represents the displacement magnitude) and Sb–Sb zig-zag chain formation. We chose nonstandard unit cell settings, $Pcmn$, $I2mm$, and $P2_1mn$ (standard setting $Pnma$, $Imm2$, and $Pmn2_1$ respectively), for the orthorhombic space groups so that the $c$ axes indicate the stacking direction.

exhibit solubility or not via synthesizing binary solid solutions, $(A,A')$MnSb$_2$. Figure 2a summarizes allowable combinations of $A$-site cations which result in a homogeneous phase. Eight of the ten pairs of $A$-site cations form a homogeneous solid solution, whereas Ba–Ca and Ba–Yb combinations exhibit phase separation, as evident from the energy-dispersive X-ray spectroscopy (EDS) mapping images shown in Fig. 2b and c. This is most likely due to the large ionic size mismatch between Ba and Ca (Yb), and the stacking arrangement of the MnSb$_4$ layers has a marginal impact as BaMnSb$_2$ and SrMnSb$_2$ are mixable despite their different stacking sequences. Such unavoidable phase separation further limits the structural design palette accessible by the solid-solution approach.

Although some binary pairs do not show extensive solubility, we successfully obtained a homogeneous phase by introducing the five elements into the $A$ site. Based on the crystallographic features given above, we chose a Ba-rich composition (Ba is 40% and the others are 15% each) aiming to create a polar high-entropy phase. We note that this composition still has a large configurational entropy ($\simeq 1.50R$; $R$ is the universal gas constant), close to the equimolar ideal case with five different cations ($\simeq 1.61R$) and higher than the four-component solid solutions with equimolar ratio ($\simeq 1.38R$). Chemical composition mapping obtained by EDS indicates homogeneous distributions of the five elements on a micrometer length scale [Fig. 2d]. The EDS measurements were also employed to investigate the material composition of the $A^5$MnSb$_2$ crystal. The chemical composition is revealed to be (Ba$_{0.38}$Sr$_{0.14}$Ca$_{0.16}$Eu$_{0.16}$Yb$_{0.16}$)MnSb$_2$, close to the nominal atomic percentages. Short-range ordering is less likely to exist because we did not observe any diffuse reflections in our selected-area electron diffraction (SAED) experiment presented below.

Having established that the $A^5$MnSb$_2$ crystal shows uniform spatial distributions in its composition, we next assess the crystal structure of this high-entropy material. Figure 3a displays the optical second harmonic generation (SHG) intensity collected from the as-grown surface of the crystal as a function of the polarization angle of the incident light. The SHG signal is restricted to zero by symmetry if a crystal is invariant under spatial inversion[43]. In $A^5$MnSb$_2$, we observed a very strong SHG signal, sharply contrasted with the extremely weak signal due to surface contribution observed in centrosymmetric YbMnSb$_2$ (see Supplementary Note 4 for detail). This indicates that the SHG signal observed in the high-entropy crystal originates from the

intrinsic effect due to the lifted inversion symmetry in the crystal structure rather than extrinsic surface contributions.

To identify the atomic arrangement of this material as well as its stacking sequence of the MnSb$_4$ layers along the $c$ axis, we conduct annular dark-field scanning transmission electron microscopy (ADF-STEM) imaging. Figure 3b, c displays the ADF-STEM image along the [010]- and [100]-zone axes, respectively. These images indicate that the MnSb$_4$ layers in the $A^5$MnSb$_2$ are directly aligned like those in (Sr/Ca/Eu/Yb)MnSb$_2$[26,39,40]. The Sb ions on the conducting Sb layers are slightly shifted and form Sb-Sb dumbbells when seen from the [010]-direction, but evenly spaced when observed from the [100]-direction as follows. The average values of $d1$ and $d2$ distances [defined in Fig. 3b] for the ADF-STEM image along the [010] direction are 2.206(17) and 2.402(7) Å, respectively, clearly indicating the dimer formation. On the other hand, the two average distances estimated for the image along the [100]-direction are very close: $d1 = 2.243(5)$ Å and $d2 = 2.210(5)$ Å. These results provide direct evidence for the Sb zig-zag chain formation in $A^5$MnSb$_2$, which is observed in (Ba/Sr/Ca/Eu)MnSb$_2$ as well[17,18,39,40]. Although the estimated shift of the Sb atoms ($\simeq 0.15$ Å) is smaller than that in BaMnSb$_2$ ($\simeq 0.30$ Å)[17], we anticipate that the Sb displacement is the origin of the spatial inversion symmetry breaking as discussed later.

From these structural features, i.e., the aligned MnSb$_4$ layers and Sb zig-zag chains, one may expect that the high-entropy phase adopts $Pcmn$ ($Pnma$ in the standard setting) structure similar to $A$MnSb$_2$ ($A$ = Sr, Ca, and Eu)[39,40]. Although $A^5$MnSb$_2$ and SrMnSb$_2$ may be very close, the $Pcmn$ structure is centrosymmetric and thus inconsistent with the SHG observed in $A^5$MnSb$_2$. Furthermore, the $00l$ ($l$: odd) reflections violating the extinction condition of space group $Pcmn$ were observed in SAED as shown in Fig. 3(d), thus ruling out the possibility of the $Pcmn$ structure. The possible occurrence of stacking faults is discussed in Supplementary Note 5. Given that $A^5$MnSb$_2$ involves high-entropy induced lattice distortions, it is likely to have lower symmetry than $Pcmn$. Note that no diffraction spot implying a superlattice larger than $\mathbf{a}_p \times \mathbf{b}_p \times 2\mathbf{c}_p$ is detected; the lattice distortion present in the high-entropy crystal should have a propagation vector $\mathbf{k} = (0, 0, 0)$, i.e., $\Gamma$-point distortion. As summarized in Table 1, four possible non-centrosymmetric structures (as well as four centrosymmetric ones, $Pcmn$, $P2_1/c$, $P2_1/n$, and $P2_1/m$) are generated from the parent $Pcmn$ structure using ISODISTORT[44] when considering $\Gamma$-point distortion, Of these, only the $P2_1mn$ ($Pmn2_1$ in standard setting) phase allows $00l$ ($l$:

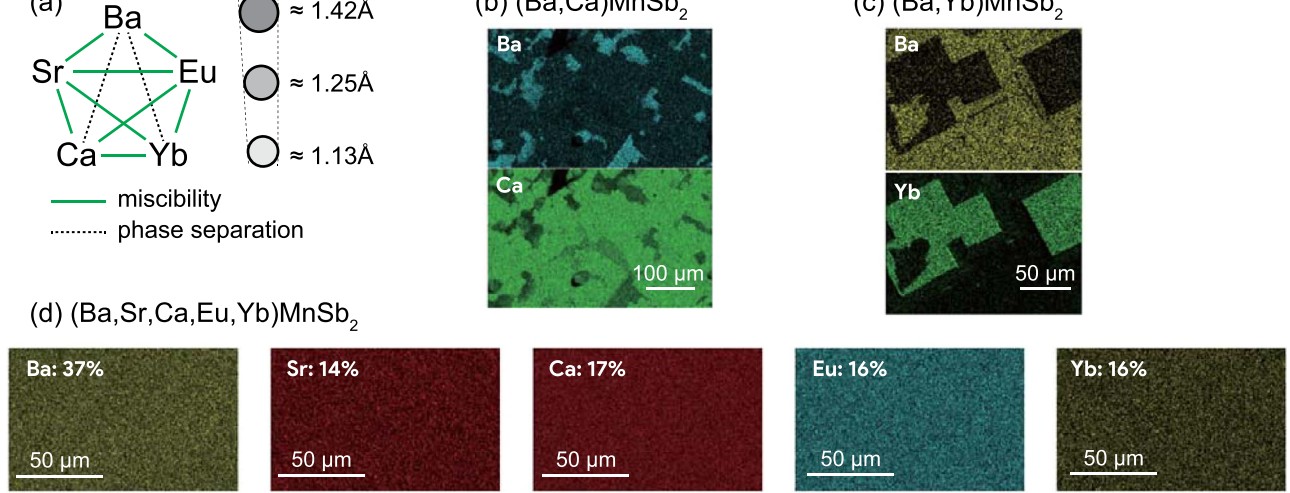

**Fig. 2 | Miscibility of the $A$-site cations in the $A$MnSb$_2$ system. a** Diagram summarizing whether ten possible binary mixings of $A$-site cations yield a homogeneous solid solution system $(A,A')$MnSb$_2$. Approximate values of Shannon's eight-coordinate ionic radii are shown for the five $A$-site cations. Sr$^{2+}$ and Eu$^{2+}$, and Ca$^{2+}$ and Yb$^{2+}$ have almost identical ionic radii to each other. EDS composition mapping for the $A$-site elements collected for **b** (Ba,Ca)MnSb$_2$, **c** (Ba,Yb)MnSb$_2$, and **d** $A^5$MnSb$_2$ samples.

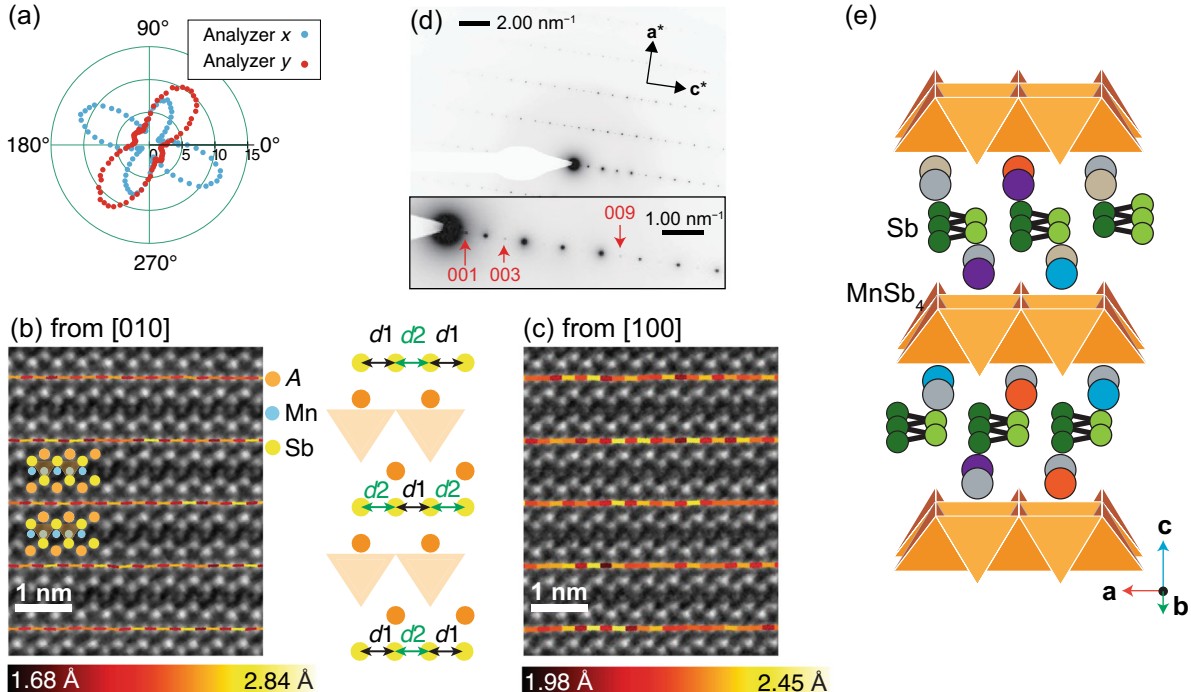

**Fig. 3 | Structural characterizations for high-entropy $A^5MnSb_2$ crystal. a** SHG intensity (radial axis in arb. units) as a function of the polarization angle of the incident light (angular axis; 0° and 90° correspond to lab $x$ and $y$ directions, respectively) measured for $A^5MnSb_2$. The sample shows 90° domains and was mounted so that the crystallographic $a/b$ and $b/a$ axes of each domain are almost parallel to lab $x$ and $y$ directions, respectively. **b** Experimental ADF-STEM image along **b** [010] and **c** [100] zone axis with a superimposed atomic structure. The lines connect the nearest Sb atoms, whose color indicates the Sb–Sb distances. The schematic between the two ADF-STEM images defines two Sb–Sb distances, $d1$ and $d2$. **d** SAED pattern along [010] zone axis, where the inset highlights the additional $00l$ reflection spots ($l$ = odd integers). The contrast is reversed to well visualize weak diffraction spots. **e** Crystal structure of the high-entropy $P2_1mn$ structure, where the random distribution of the $A$ site is represented by different colors of the $A$-site atoms.

odd) reflections experimentally observed for $A^5MnSb_2$. Our diffraction study complemented by group-theoretical analysis suggests that the high-entropy phase belongs to a noncentrosymmetric and polar space group $P2_1mn$ [Fig. 1c].

To obtain further insight into the structural features of the high entropy $A^5MnSb_2$, structural refinement based on the single crystal XRD data was carried out. Due to the X-ray diffraction limits, the $A$ site was treated as being singly occupied by Ba ions, which has a close number of electrons to the average electron count of the $A$ site, with potential vacancies. We have performed structural refinements with all the candidates including the centrosymmetric ones, but due to excessively high $R$ values, the refinements based on centrosymmetric space groups were deemed unfeasible. In contrast, the refinement was successfully performed using the four noncentrosymmetric space groups shown in Table 1. The goodness of fit indicators for the four space groups are very similar: 1.072 for $P2_12_12_1$, 1.132 for $P2_1mn$, 1.095 for $Pmc2_1$, and 1.184 for $Pc2_1n$; thus we cannot unambiguously conclude that the high-entropy crystal has $P2_1mn$ structure only based on the refinement results. However, the successful refinement with the $P2_1mn$ model supports our space group assignment based on SAED pattern and group-theoretical analysis. The statistics and crystallographic information on the refined structures are listed in Supplementary Tables 1–5.

Nevertheless, the refined $d1$ and $d2$ values ($\simeq 2.147$ and $2.234$ Å, respectively) are still somewhat different from those detected by STEM observation. To include the static Sb displacement properly, we utilize density-functional-theory-based structural relaxation with the lattice parameters fixed to the experimental values obtained by the single crystal XRD. We assumed a 100%-occupation of each of Ba, Sr, or Ca for the $A$ site. Eu and Yb were excluded to avoid the complexity of the band structure due to the contribution from f orbitals. In the case of Ba and Sr, the $P2_1mn$ structure relaxes to $Pcmn$, whereas it remained in the $P2_1mn$ symmetry when Ca occupies the $A$ site [schematically depicted in Fig. 3e]. In addition, this relaxed structure has the closest Sb-displacement values to those observed in the ADF-STEM image [Supplementary Fig. 10a]. In the following discussion, we will use these atomic coordinates obtained by assuming Ca occupation for simulating the electronic structure of the high-entropy crystal.

### Persistent Dirac dispersion in $A^5MnSb_2$

We performed angle-resolved photoemission spectroscopy (ARPES) measurements to reveal the electronic structure of the $P2_1mn$ phase. Figure 4a depicts the constant energy contour acquired by the intensity integration over the energy range from −50 to −70 meV, from which hole pockets near the Γ and X points are observed. Such an electronic structure has been identified for $SrMnSb_2$, $CaMnSb_2$, and $EuMnSb_2$, all of which adopt the $Pcmn$ ($Pnma$ in standard setting) space group[45]. The hole pocket at the X point is known to be comprised of gapped Dirac node; in this materials family, Sb displacements or inclusion of spin-orbital coupling leads to the gap opening at the X

## Table 1 | List of the possible noncentrosymmetric structures that can be created by applying Γ-point distortion to the $Pmcn$ structure

| Structure | Distortion mode | Zig-zag formation | $00l$ reflection ($l$: odd) |
|---|---|---|---|
| $P2_12_12_1$ | $\Gamma_1^-$ | Allowed | Forbidden |
| $P2_1mn$ | $\Gamma_2^-$ | Allowed | Allowed |
| $Pcm2_1$ | $\Gamma_3^-$ | Allowed | Forbidden |
| $Pc2_1n$ | $\Gamma_4^-$ | Allowed | Forbidden |

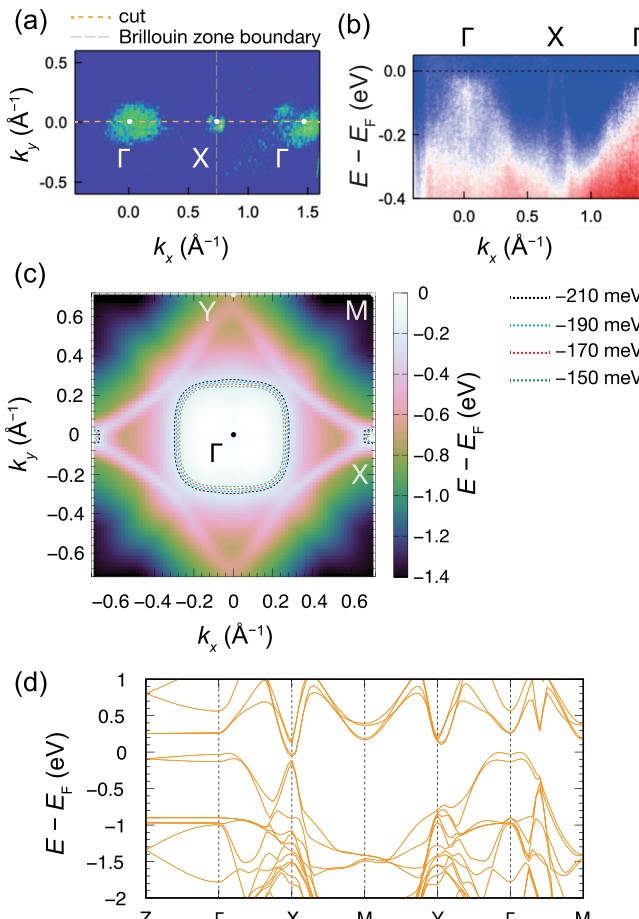

**Fig. 4 | Dirac-like dispersion in the high-entropy system. a** Constant energy contour of $A^5MnSb_2$ on the $k_x$–$k_y$ plane obtained by integrating the intensity from $E - E_F = -50$ to $-70$ meV. **b** ARPES band mapping along the Γ–X direction (cut in panel **a**). Calculated **c** constant energy contour map in $k_x$–$k_y$ plane ($k_z = 0$) and **d** electronic band structures, where Ca 100% occupation for the $A$ site is assumed. Given the hole-doped nature of the high-entropy crystal, contours below $E_F$ are highlighted in panel **c**.

point[28]. Due to the hole-doped nature, the band top of the gapped Dirac state was inaccessible by ARPES measurements, but the steep linear $E(k)$ dispersion measured along Γ–X [Fig. 4b] corroborates the presence of the gapped Dirac cone. To the best of our knowledge, $A^5MnSb_2$ is the first material possessing both high configurational entropy and linearly dispersed Dirac bands. The constant energy map in the $k_x$–$k_y$ plane and gapped Dirac-like dispersion are reasonably reproduced by the density-functional-theory (DFT) calculations [Fig. 4c, d]. Note that the DFT calculation, where Ca 100% occupation was assumed, probably overestimates the energy of the trivial band around the Γ point. The occupation of Sr or Ba leads to a downward shift of the energy of the parabolic band at the Γ point [Supplementary Figs. 10b, c]. Since $A^5MnSb_2$ contains about 40% of Ba at the $A$ site, the actual energy at the Γ point should be much lower than that drawn in Fig. 4d.

Given that $P2_1mn$ is a noncentrosymmetric and polar space group, one can expect a spin-valley locked state for $A^5MnSb_2$ as well, as in the case of $BaMnSb_2$ with space group $I2mm$. For $BaMnSb_2$, two Dirac cones intersecting near the X point were observed by ARPES[17,18]. As for $A^5MnSb_2$, unfortunately, the crystals are heavily hole-doped, and the chemical potential is too low to observe linear band crossing points. Once an electron-doped sample is synthesized, a small splitting originating from the spin-valley locking state might be experimentally observable. This spin-valley locked state should have a smaller

separation between the two valleys due to smaller orthorhombic distortion in $A^5MnSb_2$.

## Non-trivial Berry phase and quasi-3D Fermi surface

After demonstrating that $A^5MnSb_2$ hosts Dirac dispersions, we next study its transport behavior. Magnetotransport properties were measured for the same crystal used for the ARPES experiments. Figure 5a shows the in-plane longitudinal resistivity ($\rho_{xx}$) and Hall resistivity ($\rho_{xy}$) as a function of the magnetic field ($B$) at various temperatures. Given that the mobility from the Fermi pocket at Γ point is extremely lower than that of X point (Supplementary Note 1), linear $\rho_{xy}$–$B$ curves, which can be described by a single-band model, indicate that the transport of this crystal is dominated by the linear Dirac bands found near the X point. The Hall coefficient $R_H$ extracted from the slope of $\rho_{xy}$–$B$ curve (in the range of 0–10 T) remains a positive constant value up to 50 K, indicating that a hole-like band dominates the transport property of the material. This is consistent with the hole-doped nature of this crystal revealed by the ARPES measurements. We next estimate the carrier density ($n_H$) and Hall mobility ($\mu$) at 1.7 K using the relations $n = 1/eR_H$ and $\mu = R_H/\rho_{xx}(B = 0)$, respectively, where $e$ is the electron charge. The estimated values are $n_H = 8.3 \times 10^{18}$ cm$^{-3}$ and $\mu = 5.9 \times 10^3$ cm$^2$/Vs. The mobility of $A^5MnSb_2$ is comparable to those of the parent $AMnSb_2$ series[23,25,46], showing that transport of the charge carriers is not hindered by the presence of configurational disorder at the $A$ site.

Such high mobility allows for detecting Shubnikov–de Haas (SdH) oscillations on $A^5MnSb_2$ crystals, as evident in Fig. 5a. We also observed the Zeeman splitting at lower temperatures than 30 K; the extraction of Landé $g$-factor is given in Supplementary Note 1. To understand the nature of the SdH oscillations, we analyzed the $\rho_{xx}$ data by taking the second derivative [Fig. 5b] and then performed the fast Fourier transform (FFT) analyses. The FFT spectra obtained at various temperatures are plotted in the inset of Fig. 5b, composed of only a single frequency of $F_\alpha = 79$ T. Note that the FFT peaks appearing at around 160 and 240 T are the second ($F_{2\alpha}$) and third harmonic frequencies ($F_{3\alpha}$), respectively. The obtained oscillational frequency is between those reported for $AMnSb_2$ and $AMnBi_2$ Dirac semimetals[47–51]; in other words, the small $F_\alpha$ value is in the frequency range generally expected for topological semimetals with the Dirac node being near the Fermi level. Onsager relation, $F_\alpha = \frac{\Phi_0}{2\pi^2} S$, directly links $F_\alpha$ to the extremal Fermi surface cross-section $S$, where $\Phi_0$ is the flux quantum. Accordingly, $F_\alpha$ corresponds to the cross-sectional area of 0.75 nm$^{-2}$ and is consistent with the ARPES-extracted pocket size along $k_x$ direction [0.096 Å$^{-1}$, Fig. 4b], which leads to a cross-sectional area of 0.72 nm$^{-2}$ when assuming a circular shape. Moreover, the larger $F_\alpha$ value of $A^5MnSb_2$ than the pristine $AMnSb_2$ indicates a larger $S$, implying that the Fermi level is slightly off the Dirac node, in good agreement with the ARPES measurements.

From the temperature-dependent FFT amplitude [the inset of Fig. 5b], we can obtain effective mass, $m^*$ via the fit to the temperature damping factor $R_T$, which can be expressed as follows according to Lifshitz–Kosevich (LK) theory[52]:

$$R_T = \frac{\frac{2\pi^2 m^* k_B T}{\hbar e B}}{\sinh \frac{2\pi^2 m^* k_B T}{\hbar e B}}, \tag{1}$$

where $k_B$ is Boltzmann's constant and $\hbar$ is the reduced Planck constant. As for the value of $B$ for fitting, we used the average of the minimum and maximum applied magnetic fields (0 and 35 T, respectively). Our LK fitting shown in Supplementary Fig. 4 yields an $m^*$ of 0.51$m_0$, where $m_0$ is the mass of a stationary electron. Such a heavy effective mass was also reported for other Dirac materials[51,53], implying a massive Dirac fermion in the high-entropy crystal (see also Supplementary Note 2).

A non-trivial Berry phase is an additional evidence for topological semimetals[54,55], which can be assessed via constructing a Landau level

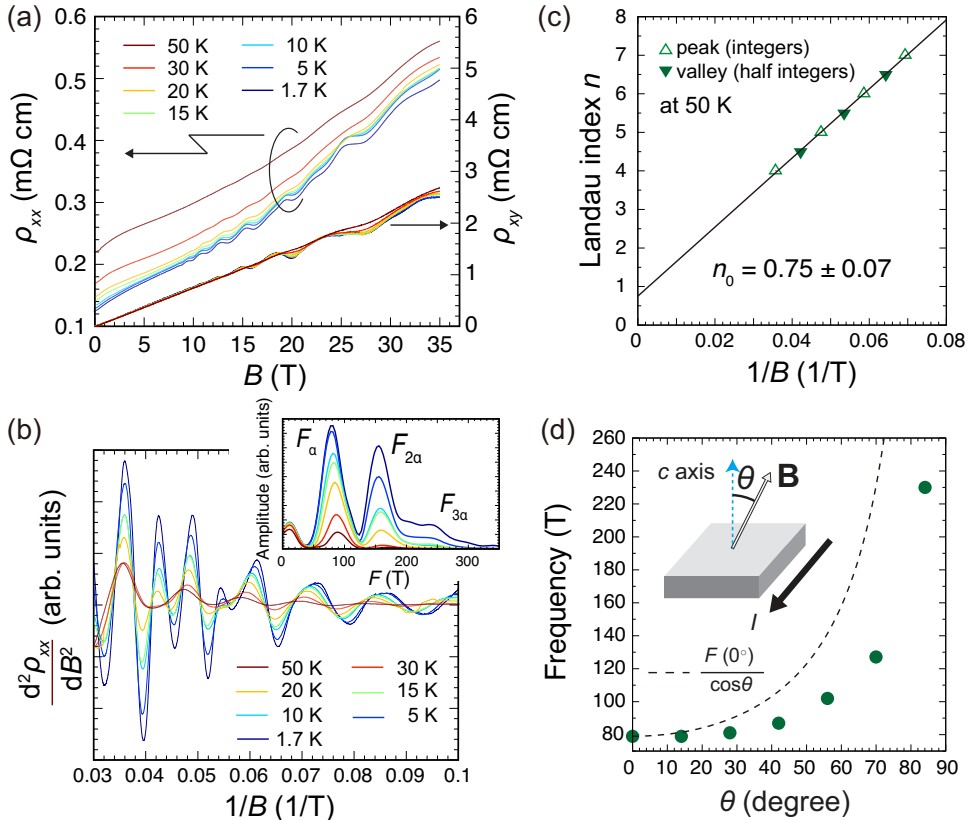

**Fig. 5 | Magnetotransport measurements for $A^5$MnSb$_2$. a** In-plane electrical resistivity $\rho_{xx}$ and Hall resistivity $\rho_{xy}$ of $A^5$MnSb$_2$ as a function of magnetic field $B$ at various temperatures (1.7, 5, 10, 15, 20, 30, and 50 K). **b** The second derivative of $\rho_{xx}$ as a function of $1/B$. The inset shows the temperature evolution of the FFT spectra of $\frac{d^2\rho_{xx}}{dB^2}$. **c** Landau fan diagram obtained from $\frac{d^2\rho_{xx}}{dB^2}$ at 50 K, where the integers are assigned to the maxima of the second-derivative curve. The error of $n_0$ is obtained from the least square fit to the diagram, representing the uncertainties of the fitted intercept. **d** Angular dependence of the SdH frequency ($F_\alpha$) extracted from the $\rho_{xx}$ data at 1.7 K. The dashed curve expresses $F_\alpha(0°)/\cos\theta$.

(LL) fan diagram based on the oscillatory part of the LK formula[56],

$$\Delta\rho \propto \cos 2\pi\left[\frac{F_\alpha}{B} + \left(\frac{1}{2} - \frac{\phi_B}{2\pi}\right) - \delta\right], \qquad (2)$$

where $\phi_B$ and $\delta$ are the Berry phase and the phase shift depending on the dimensionality of the Fermi surface, respectively. The $\delta$ is equal to 0 for Dirac systems having 2D Fermi surface like $A$MnSb$_2$. We use the $\frac{d^2\rho_{xx}}{dB^2}$ curve [Fig. 5b], which is out-of-phase with $\rho_{xx}$, to build the LL fan diagram, where integers (half-integers) are assigned to the peaks (valleys). Here, the 50-K data is used to circumvent the effect of Zeeman splitting causing peak splits on the oscillatory component, which is prominent at lower temperatures. As shown in Fig. 5c, the linear least square fitting to the diagram and the extrapolation yields the intercept $n_0$ of 0.75(7), which is clearly larger than the expected value of 0.5 for two-dimensional Dirac systems with a Berry phase of $\pi$. Although the difference in chemical potential is a possible explanation for the deviation as studied previously[17], it is most likely due to the quasi-3D character of the Dirac bands in the high entropy phase as we discuss below. The overall results presented above conclusively demonstrate that Dirac fermion persists in the high-entropy crystal.

Further, we have analyzed the angular dependence of the SdH oscillation frequency [Supplementary Fig. 5]. The experimental setup is shown in the inset of Fig. 5d. The LL fan diagram obtained from the high field-angle data also results in a deviated intercept from 0.5, 0.62(9) [Supplementary Fig. 5d]. If the Fermi surface is of strong 2D nature, which is the case in the pristine $A$MnX$_2$ ($A$ = Ba, Sr, Ca, Eu, and Yb; $X$ = Sb and Bi), $F_\alpha$ is expected to increase with $\theta$, and the $F_\alpha$-vs-$\theta$ plot can be well fitted with the $1/\cos\theta$ curve. As shown in Fig. 5d,

on the contrary, the angular dependence of $F_\alpha$ obtained for $A^5$MnSb$_2$ displays a significant deviation from the $1/\cos\theta$ curve, and the SdH oscillations are still observable even at $\theta \simeq 90°$. This indicates that the Fermi surface responsible for the SdH oscillations in $A^5$MnSb$_2$ supports closed cyclotron orbits in the range of $0° \le \theta \le 90°$, implying a highly-anisotropic character (i.e., quasi-3D) of the morphology instead of strong 2D. This accounts for the small deviation of $n_0$ from the expected value of 0.5 mentioned above. In the case of materials with a 3D Fermi surface, the phase shift $\delta$ of Eq. (2) takes the value of $\pm 1/8$, and thus the $n_0$ is expected to be 0.625 or 0.375. Our observed deviation of $n_0$ from 0.5 is clearly consistent with this expectation.

Given that the $P2_1mn$ structure of $A^5$MnSb$_2$ also has a layered structure, it should generate a strong-2D electronic structure as seen in the parent compounds. What leads $A^5$MnSb$_2$ to show quasi-3D nature in its electronic structure? Our theoretical calculations find that the local structure distortions induced by the configuration entropy plays a key role in generating such a quasi-3D electronic structure, as to be discussed below. The DFT-relaxed $P2_1mn$ structure possesses almost flat Sb layers, as illustrated in Fig. 6a, and does not reproduce such a quasi-3D nature of the Fermi surface [Fig. 6b]. Considering the strong lattice distortion in the high-entropy phase, we here introduce the rumpling of Sb atoms to the relaxed structure, which is totally symmetric (i.e., transforming as the irreducible representation $\Gamma_1$) and thus appears intrinsically in the Sb layers without breaking the $P2_1mn$ symmetry. The rumpling distortion consists of anit-polar displacements of adjacent Sb atoms along the $c$ direction [Fig. 6a] and is analogous to the buckling of graphene[57]. The $k_y$–$k_z$ contour map [Fig. 6c] shows the Fermi surface at $k_x = 0.5$ computed for the buckled

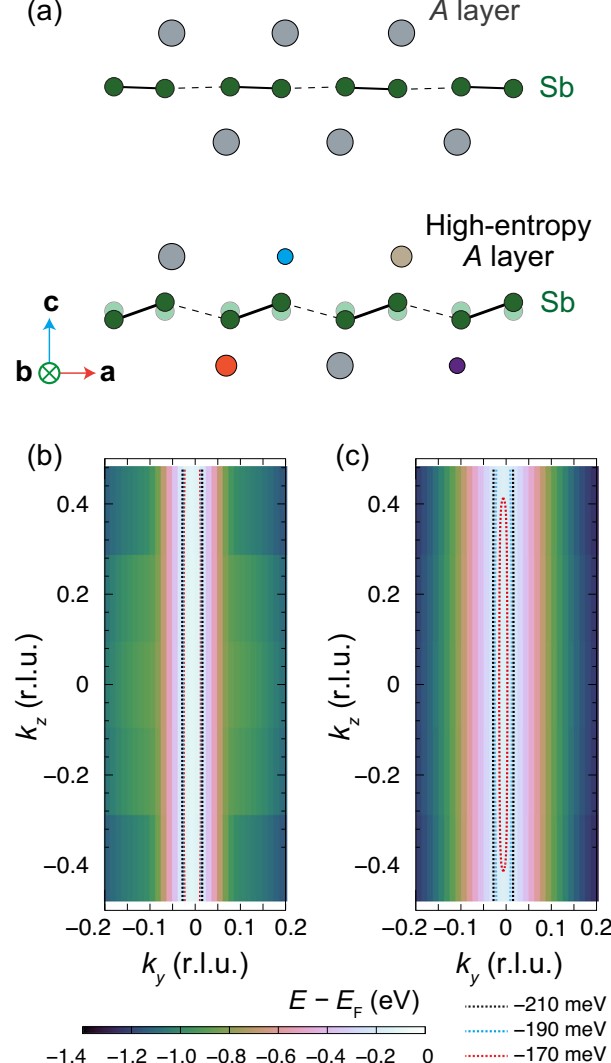

**Fig. 6 | Dimensionality change of the Fermi morphology in $A^5$MnSb$_2$ caused by local structural distortion. a** The schematic showing pristine and rumpled Sb atoms at the interface between Sb and $A$ layers. Fermi surface contour maps in $k_y$–$k_z$ plane at $k_x = 0.5$ calculated for **b** relaxed and **c** rumpled $P2_1mn$ structures. Given the hole-doped nature of the high-entropy crystal, contours below $E_F$ are highlighted.

coupling between them. We anticipate that the indirect distortion induced in the MnSb$_4$ layers also contributes to the formation of a quasi-3D Fermi surface.

## Discussion

Structures with aligned MnSb$_4$ layers keep the atomic position of Sb crystallographically identical even under orthorhombic strain, leading to the centrosymmetric *Pcmn* structure [Fig. 1c]. Although $A^5$MnSb$_2$ also adopts the aligned MnSb$_4$ sequence, this phase crystallizes in the polar space group *P2$_1$mn* with macroscopic polarization. This is probably because of inequivalent coordination around each Sb atom; on average, the coordination environment at each Sb site is the same due to the random distribution of the $A$ cation but distinct at a local scale. Such a local inhomogeneity creates crystallographically inequivalent Sb sites so that the Sb displacements have different magnitudes in the same manner as in the anti-aligned structure [Fig. 1c]. Consequently, the anti-parallel electric dipoles induced by the Sb displacements no longer canceled out each other, resulting in net macroscopic polarization. This scenario is similar to that for the emergence of improper ferroelectric polarization[61] and can explain why $A^5$MnSb$_2$ favors the polar structure despite the aligned stacking of MnSb$_4$ layers.

The layered structure of the $A$MnSb$_2$ series enables the coexistence of high mobility and configurational disorder in $A^5$MnSb$_2$, where the Sb slabs with orthorhombic distortion host Dirac fermions, whereas $A$-MnSb$_4$ slabs accommodate randomness. Owing to the robustness of Dirac state against a certain magnitude of disorder[62], we think that the transport of Dirac materials is not severely affected by disorder up to a critical point; the $A^5$MnSb$_2$ case may fall into this regime due to the layered nature.

Occupations of different cations in a single sublattice generally provide local distortions[59,60], leading to strong phonon scattering and, thus, low thermal conductivity[63,64]. Change in the phonon frequency was detected by Raman spectroscopy (Supplementary Note 4). Therefore, high-entropy $A$MnSb$_2$ series might be ideal materials for thermoelectric applications. Moreover, the pristine $A$MnSb$_2$ series itself is proposed as a semimetal family for high thermoelectric performance owing to their electronic structures near the Fermi level[65]. Overall, we believe that measuring the thermoelectric performance of a series of $A$MnSb$_2$ alloys is an interesting direction for future investigations.

We anticipate that the high-entropy strategy to create a structure inaccessible by the binary solid-solution approach is generally applicable. The mechanism behind the emergence of a new crystal structure is a compromise among crystalline phases with distinct structures and/or symmetries. In the case of $A^5$MnSb$_2$, there are three different polymorphs adopted by the $A$MnSb$_2$ series (*P4/nmm*, *Pcmn*, and *I2mm*); two of them prefer aligned MnSb$_4$ layers with centrosymmetry, while the other crystallizes in a noncentrosymmetric structure with anti-aligned MnSb$_4$ layers. The resulting high-entropy structure possesses aligned MnSb$_4$ layers and a noncentrosymmetric structure, which can be viewed as a compromising consequence of the distinct structural features. Other examples of such structural compromises reported elsewhere[32,37] are provided in Supplementary Note 6.

Therefore, we can generally anticipate that when different structures whose chemical compositions are akin to each other are identified, one can mix them to achieve a new structure by utilizing this compromise mechanism. In this sense, we believe that this strategy is widely applicable. Of course, such a compromise also plays a crucial role in the cases of binary solid solutions; however, an insufficient configurational entropy often ends up with a phase separation as observed in (Ba,Ca)MnSb$_2$ and (Ba,Yb)MnSb$_2$ [Fig. 2b, c]. Thus, high configurational entropy needs to be employed to merge distinct structures into a homogeneous new structural phase.

To summarize, we have successfully synthesized high-entropy crystals of (Ba$_{0.38}$Sr$_{0.14}$Ca$_{0.16}$Eu$_{0.16}$Yb$_{0.16}$)MnSb$_2$ even though some of the binary pairs do not show extensive solubility. Through

structure, where a closed quasi-3D pocket emerges in striking contrast to those calculated for non-buckled structure [Fig. 6b]. The ratio between the in-plane and out-of-plane cross-section areas ($S_{xy}$ and $S_z$, respectively) is calculated as $S_z/S_{xy} = 3.3$. In terms of the Onsager relation, this ratio matches with the SdH-frequency ratio, $F_\alpha(90°)/F_\alpha(0°)$. Since our angular-dependent magnetotransport measurements [Fig. 5d] give the estimation of the frequency ratio as $F_\alpha(90°)/F_\alpha(0°) \simeq F_\alpha(84°)/F_\alpha(0°) = 2.9$, the simplified DFT model, to some extent, captures the experimental results. 2D-to-3D reconstruction of the Fermi surface has been observed across the charge-density-wave transition in other materials[58]. In $A^5$MnSb$_2$, however, the dimensionality change of the Fermi morphology is probably caused by the local rumpling distortion that stems from high-entropy induced atomic-size/mass/bonding-state misfit[59,60] and keeps the crystal symmetry invariant.

Since the high-entropy $A$-cation layers are also adjacent to MnSb$_4$ tetrahedral layers, we expect that such out-of-plane displacive distortion also exists in the MnSb$_4$ layers (specifically, Sb layers and Mn layers). The out-of-plane displacement within MnSb$_4$ layer would assist the electron tunneling across the adjacent Sb$_4$ layers, increasing the

experimental and group theoretical approaches, we demonstrate that the high-entropy concept allows access to an otherwise unobtainable crystal symmetry, polar $P2_1mn$ in this example, which does not correspond to any of the parent $A$MnSb$_2$ structures. In addition to crystal structure, local disorder also affects electronic characteristic in reciprocal space, i.e., the evolution of the Fermi-surface dimensionality from 2D to quasi-3D, as shown by our analysis of the SdH oscillations, which is supported by first-principles calculations. Given that the crystal and/or electronic structures govern the physical properties of solids, our finding implies that the high-entropy approach is a promising way to unlock topological characteristics.

## Methods

### Experimental procedures

Single crystals of $A^5$MnSb$_2$ were synthesized by the Sb self-flux method. Ba, Sr, Ca, Eu, Yb, Mn, and Sb elements were used as the starting materials. They were mixed in the molar ratio of Ba:Sr:Ca:Eu:Yb:Mn:Sb = 0.4:0.15:0.15:0.15:0.15:1:6, put into an alumina crucible, covered with quartz wool, and sealed in an evacuated quartz tube. After being heated at 1000 ℃ for 6 h, the ampule was cooled to 700 ℃ at a rate of 3 ℃/h. Then, plate-like single crystals with a typical size of 2 mm × 2 mm × 0.5 mm were separated from the excess Sb flux by centrifuging at 700 ℃. The compositions of the grown crystals were characterized using EDS. Several single crystals were picked up and examined in Rigaku Synergy-S single-crystal X-ray diffractometer equipped with Mo radiation ($\lambda_{K\alpha}$ = 0.71073 Å) to obtain the structure and crystal facet information. The crystal was measured with an exposure time of 10 s and a scanning $2\theta$ width of 0.5° at room temperature. The data was processed in the CRYSALIS software, and the structural refinements were conducted with the SHELXTL package using direct methods and refined by full-matrix least-squares on $F^2$.

The optical SHG measurements were carried out on the as-grown surface of the crystal in reflection geometry at room temperature. A pulsed fundamental beam generated by a regeneratively amplified Ti:Sapphire laser system ($\lambda$ = 800 nm, repetition rate of 1 kHz) was used as the light source. The polarization of the fundamental beam was controlled by a half-wave plate, and the second harmonic signal generated through the nonlinear optical process inside the sample was detected by a photo-multiplier tube after passing through a polarization analyzer. The SHG intensity was plotted as a function of the polarization direction of the fundamental beam.

The STEM analysis was conducted on a sample prepared using Thermo Fisher Helios NanoLab Dual-Beam Focused Ion Beam. One cross-sectional lamella was lifted out on a structural domain to observe the crystal along both [100] and [010] directions. The atomic resolution ADF-STEM experiments were performed using dual spherical aberration corrected Thermo Fisher Titan3 G2 S/TEM at 300 kV accelerating voltage. Thermo Fisher Talos F200X at 200 kV accelerating voltage was used to acquire the SAED data. The SAED patterns were acquired using a circular aperture that spans about 800-nm projected diameter. Multiple SAED patterns were obtained from different regions, and the results are consistent with each other.

The ARPES measurements were performed at Beamline 10.0.1, Advanced Light Source, Lawrence Berkeley National Laboratory. By cleaving the $A^5$MnSb$_2$ crystal, we first achieved its pristine (001) surface at 25 K. The base vacuum of the ARPES chamber is better than ~5 × 10⁻¹¹ mbar. A hemispherical Scienta R4000 analyzer was used in our ARPES measurements. The energy and angle resolutions were set to ~15 meV and ~0.1°, respectively. The photon energy used in ARPES measurements is 53 eV and the spot size of the beam is ~100 × 100 μm². The linear horizontal polarization of incident light is used.

The magnetotransport measurements were performed using Stanford Research Systems 580 current sources and 860 lock-in amplifiers. Fields up to 35 T were provided by a water-cooled resistive magnet at the National High Magnetic Field Laboratory in Tallahassee,

FL USA. The standard four-probe technique was employed for the longitudinal resistivity $\rho_{xx}$ and Hall resistivity $\rho_{xy}$ measurements, where a small DC of 1 mA was applied. A cryostat was fitted with a variable temperature insert to provide stable measurement temperatures that were confirmed by a Cernox thermometer located 1.5 cm from the samples. The crystals were mounted to a sample platform that allowed for in-situ rotation with respect to the applied magnetic field to explore the angular dependence of the Fermi surface.

### Computational details

First-principles calculations were performed using the plane-wave (PW) pseudopotential method implemented in the QUANTUM-ESPRESSO suite[66–68] with the Perdew-Burke-Ernzerhof (PBE)[69] parametrization of the generalized gradient approximation (GGA) and with pseudopotentials from the PSEUDO DOJO library[70]. Calculations were performed at the GGA + $U$[71–73] level using Löwdin-orthogonalized atomic projectors[74] with the Hubbard $U$ correction being applied to Mn, whose magnetic moments followed the C-type antiferromagnetic ordering. Hubbard parameters were obtained for CaMnSb$_2$, SrMnSb$_2$, and BaMnSb$_2$ in the $P4/nmm$ structure using density functional perturbation theory[75,76] employing the procedure outlined in ref. 77 to compute $U$ parameters nonempirically.

The computed values of the Hubbard parameters for the three compositions were $U_{Mn,Ca}$ = 4.4 eV, $U_{Mn,Sr}$ = 4.7 eV, and $U_{Mn,Ba}$ = 4.6 eV. The value of $U_{Mn}$ = 4.6 eV was used for all subsequent calculations with supercells of 16 atoms. The kinetic energy cutoffs for the wavefunctions and charge density were 90 Ry and 1080 Ry, respectively. Self-consistent-field calculations were performed while sampling the Brillouin zone with an origin-centered (Γ-centered) 10 × 10 × 2 Monkhorst–Pack grid until energy converged within 10⁻¹⁰ Ry. In subsequent non-self-consistent field calculations, the Brillouin zone was sampled with a Γ-centered 20 × 20 × 4 Monkhorst–Pack grid.

To reduce the cost of calculations, it was assumed that the $A$-site of the $A^5$MnSb$_2$ crystal was occupied entirely by Ca, Sr, or Ba rather than the random distribution of the five elements. This simplified model is expected to still capture the electronic structure near the Fermi level as the bands in question arise from the hybridization of the Mn 3d and Sb 5p states with little contribution from the $A$-site elements. To produce the contour plots of the electron energy, the non-self-consistent-field calculations were repeated with a dense Γ-centered grid of 80 × 80 × 5 points. Calculations were performed for the $P2_1mn$ structure varying the Sb–Mn bond lengths from 3.15 to 3.35 Å.

## Data availability

The SHG and transport data generated in this study are provided in the Source data file. All EDS, ARPES, STEM data and DFT calculations needed to evaluate the conclusions in the paper are present in the paper and/or the Supplementary Information. Additional data related to this paper are available from the authors upon request. Source data are provided with this paper.

## Code availability

DFT calculations in this work were performed using open-source code QUANTUM-ESPRESSO suite (https://www.quantum-espresso.org).

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

## Acknowledgements

The major material synthesis and transport measurement efforts are supported by the US National Science Foundation (NSF) under grant DMR-2211327. The group-theoretical structural analyses, data analyses, and partial DFT calculations are supported by the Pennsylvania State University Two-Dimensional Crystal Consortium–Materials Innovation Platform (2DCC-MIP), which is sponsored by NSF Cooperative Agreement No. DMR-2039351. The initial material synthesis, TEM work, SHG experiment, and partial DFT calculations are supported by NSF through the Pennsylvania State University Materials Research Science and Engineering Center (MRSEC) DMR-2011839 (2020–2026). The AREPS measurements are supported by the Penn State MRSEC for Nanoscale Science (DMR-2011839). The ARPES facility at the Advanced Light Source was supported by the Office of Basic Energy Sciences, the U.S. Department of Energy (U.S. DOE-BES) under Contract No. DE-AC02-05CH11231. The optical characterizations are also supported by the NSF DMR-2210933. The work at the National High Magnetic Field Laboratory is supported by the NSF Cooperative Agreement No. DMR-1644779 and the State of Florida. The work at Michigan State University is supported by the U.S. DOE-BES under Contract No. DE-SC0023648. Partial computation was carried out using the computer resource offered under the category of General Projects by Research Institute for Information Technology, Kyushu University.

## Author contributions

Z.Q.M. conceived the project. A.L., S.Y. and L.Min performed the single crystal growth. A.L., S.Y., L.Min and Y.G. conducted the compositional analyses. S.Y. and V.G. carried out the optical SHG measurements. S.S. and V.G. performed Raman spectroscopy. S.V.G.A., L.Miao and N.A. did STEM experiments. J.G.J. and W.X. measured and analyzed the single crystal XRD pattern. S.Y. performed the group-theoretical structural analyses. H.Y. and C.-Z.C. conducted ARPES experiments. S.H.L., Y.G. and D.G. collected the transport data under DC magnetic fields. A.L., S.Y., S.H.L., Y.G. and Z.Q.M. analyzed the transport data. S.Y., F.M.S.V. and I.D. performed the DFT calculations. S.Y. and Z.Q.M. drafted the manuscript with contributions from all co-authors.

## Competing interests

The authors declare no competing interests.
