## [Peer Review File · Nature Communications]

High-Entropy Engineering of the Crystal and Electronic Structures in a Dirac MaterialReviewers' Comments:

Reviewer #1:

Remarks to the Author:

The authors present a comprehensive study on a high-entropy alloy of A_5MnSb_2 . They are able to successfully grow the material. By various structural microscopies, the crystal structure is determined. ARPES measurements are further carried out and the band structure, consisting of a hole pocket at the gamma point and a linear dispersion around the X point, is mapped out. The band structure is confirmed by DFT calculations. Quantum oscillations reveal a nontrivial Berry phase of π , suggesting a Dirac band. The field angle dependence of the oscillation frequency indicates a 3D Fermi surface. The experiments were carefully carried out and data are of high quality. The idea of high-entropy engineering is novel. The fact that the carrier mobility remains high in spite of disorder is surprising and welcome. Overall, the manuscript is convincing. My biggest concern is about identification of the Dirac band by electrical transport.

Both ARPES and DFT calculations indicate that the hole pocket at the gamma point is much larger than the Dirac pocket at the X point. It is natural to believe that this band will dominate the transport, unless the mobility of this band is extremely low. It would be nice to see an estimation of the upper limit of the mobility, based on the relative sizes of two pockets. It is stated on page 5 that temperature-independent linear ρ_{xy} -B curves indicate that the transport of this crystal is dominated by the linear Dirac bands. Please elaborate the reasoning behind this statement.

Determination of the Berry phase by quantum oscillations can be tricky and susceptible to errors, when multiple frequencies or Zeeman splitting are involved, or the low Landau level indices are not available. The more Landau level indices in the Landau fan diagram, the more reliable the fitted intercept would be. In Fig. 5b, at least a peak and a valley for $1/B > 0.06$ are well resolved, but these valuable data points are not included in the Landau plot. Please include them in fitting. Moreover, can the phase analysis be performed for oscillations at high field angles where the oscillations are clean, as seen in Fig. S4b? Such analysis would certainly strengthen the arguments.

How does the oscillation frequency compare to the pocket size seen in the ARPES? An agreement between two may provide a consistency check for the Dirac band.

A cyclotron mass is obtained from the temperature dependence of the oscillation amplitude. It is rather large, $0.5m_0$. The authors argue that the Dirac node may be massive (By the way, a reference needs to be provided in the manuscript). But, the dispersion of this band seems quite linear at the Fermi level according to the ARPES data. How can a steep linear dispersion generate such a large cyclotron mass? Is it possible that the oscillation is in fact from the large trivial hole pocket?

Reviewer #2:

Remarks to the Author:

This study introduces an innovative approach to create materials with relativistic electronic states, especially Dirac/Weyl semimetals, which are pivotal in current condensed matter physics and materials science. It demonstrates that by harnessing high configuration entropy, the $AMnSb_2$ Dirac material family can be modified to yield a unique polar structure with exceptional properties absent in the original compounds. This process results in the development of an entropy-stabilized high-entropy Dirac material, maintaining linear band dispersion despite lattice irregularities. Additionally, the study uncovers a remarkable transformation from two-dimensional Dirac states to highly anisotropic three-dimensional Dirac states in the high-entropy phase, substantiated by experimental data and theoretical models. As such, I would like to recommend accepting the article for publication after the authors address the following comments:

1. Phonon behaviors of the AMnSb₂ series are not discussed in the manuscript. Could the authors provide comments on or perform theoretical calculations to enhance the discussion?
2. Ideally, achieving a homogeneous solid solution is challenging in high-entropy materials. Can the authors comment on whether they observed any form of short-range order in their experiment?
3. High entropy is associated with intrinsic structural disorder. How does this type of disorder affect the transport properties of the Dirac material?

Reviewer #3:

Remarks to the Author:

In this manuscript, Laha and coauthors report the single crystal growth and characterization of a ternary antimonide AMnSb₂, where high configurational entropy was achieved on the cationic A site by mixing five different cations. Using a combination of local and bulk probes they identify a new, non-centrosymmetric crystal structure that is not found in any of the parent, single cation AMnSb₂ materials. Based on angle-resolved photoemission spectroscopy and quantum oscillation data they show that the high entropy AMnSb₂ maintains a well-defined Fermi surface dominated by linear Dirac bands despite the high degree of disorder on the A site. Moreover, they claim that the dimensionality of the Fermi surface is found to change to 3D as compared to 2D in the parent, single cation AMnSb₂.

The work that Laha and coauthors present is conceptually well designed, and the suite of various techniques used in this work is novel for high entropy materials. In particular, the comparison of the high entropy AMnSb₂ with the permutations of binary solid solution (A,A')MnSb₂ is particularly thorough and shows that entropy is the driving force for stabilizing the new crystal structure observed for the high entropy AMnSb₂. Furthermore, the observation of quantum oscillations up to a temperature of 50 K in such a disordered material is remarkable and shows the high quality of the crystals.

However, we have three significant concerns with the present work. First, the evidence in support of the main claim of realizing new structures by high entropy alloying, the determination of the crystal structure and in particular the breaking of inversion symmetry is not convincing. Second, the design criteria applied here and the general applicability of the high entropy alloying approach to generate new structures is not discussed well in the paper. Third, the claimed distortions to the Sb chains in this material are in direct contradiction with the observed quantum oscillations and moreover the evidence for the change in dimensionality is not convincing. For these reasons, we cannot recommend this paper for publication in Nature Communications in its current form. If properly addressed, these aspects would increase the impact of the paper considerably.

If the authors could please reply to the following comments.

- 1) The observation of "a significant signal" in second harmonic generation (SHG) is taken as key evidence for inversion symmetry breaking without providing details of what a significant signal is. In particular, the authors do not describe how they disentangle bulk and surface contributions to the signal, which is crucial as surfaces also break inversion symmetry and SHG is surface sensitive. We suggest the authors improve the discussion of SHG data and in addition provide measurements on a reference sample with inversion symmetry. Measuring one of the centrosymmetric, single cation AMnSb₂ materials for example lends itself to a straightforward comparison with the high entropy AMnSb₂.
- 2) Based on the observation of (00l) l=odd reflections in the SAED pattern, which are forbidden in the centrosymmetric space groups of the single cation AMnSb₂ materials, the authors perform group theoretical analysis to derive the polar space group of the high entropy AMnSb₂. However, it is known

from related materials with the same space group (Pnma, #62), that stacking faults can give rise to these otherwise forbidden reflections, e.g. in Phys. Rev. B 54, 13587 (1996). What efforts have the authors taken to exclude stacking faults or other reasons as the origin of the (00l) l=odd reflections?

3) In the structure determination, the authors employ a range of techniques, which is commendable. However, they are also conflating local and average structure. It is well known in high entropy materials that very high levels of local distortions, due to their random nature, are not necessarily representative of the average crystal structure. In terms of establishing the true average space group for this material, we are not convinced by the authors claims that single crystal x-ray diffraction is an unsuitable method here (or even, for that matter, powder x-ray diffraction). At a minimum, the authors should present goodness of fit parameters across all the candidate space groups (including the centrosymmetric options) and show that they are not distinguishable on that basis.

4) Related to the previous questions, can the authors describe how representative for the whole sample that SAED pattern is? In the methods section, it is stated the one lamella was used for STEM but is there any variation within that lamella or between different lamellae?

5) The introduction suggests that the high entropy alloying approach can be used to overcome limitations of traditional (binary) solid solution approaches and create new crystal structures. However, it is unclear to us whether this approach is generally applicable or if it is coincidence that it works in AMnSb2. To this end, it would help if the authors could clarify the general applicability of the approach and allude to the underlying mechanism of creating new crystal structure by high entropy alloying, if it exists.

6) Related to the previous question, the design principle employed here is not explained. High entropy materials are most frequently taken to be equimolar. Can the authors explain why they chose the composition with 40% Ba over for example equimolar concentration (20%) for all cations? The actual magnitude of configurational entropy is significantly reduced from the equimolar limit.

7) Regarding the 2D-3D transition: Can the authors please give a more detailed explanation for the change in Fermi surface dimensionality upon high entropy alloying? We are not convinced that the small "rumpling" of the Sb layer gives rise to a significant out of plane dispersion. Naively, one would assume that the interlayer coupling between neighboring Sb layers has to increase significantly in order to obtain a notable out of plane dispersion. However, the rumpling assumed in the calculations seems to be rather small, so I do not understand how this would create the 3D Fermi surface. In addition, Fig. 6c in the way it is presented shows that the (presumably) valence band is still highly 2D with only a very small 3D component. Furthermore, such "rumpling", random displacements in the Sb chains, would seem to be in direct contradiction with the observed quantum oscillations. Photon energy dependent ARPES data would be much stronger evidence of 3D vs 2D character – have the authors performed this measurement?

Additional comments:

8) Regarding the abstract: The exact composition of the material should be provided in the abstract. As written, it is implied that the material is equimolar, which it is not.

9) Regarding the STEM data: the authors claim that SAED has "superior detection sensitivity to conventional x-ray based techniques". Can they elaborate more on what feature refer to in terms of superior detection sensitivity and why SAED is better than X-ray. Including a reference would be helpful here.

10) Regarding SC-XRD: For single crystal XRD data it is common practice to include the results table, including space group, number of reflections, R values and goodness of fit for the refinements etc. Could the authors please provide these data in the supplementary material?

11) Regarding SC-XRD: Based on what argument is Eu used as the cation in the refinement? Based on the composition from EDS the average electron count on the A site should be ~ 51 (much closer to Ba than to Eu), which would result in an occupation of the A site of only 0.8. Again, these results should be reported in the supplementary. In general, the choice of cations for various modellings (XRD, DFT) appears to be rather inconsistent. Could the author please explain why they chose certain cations for certain models?

12) Regarding SC-XRD: It is not clear to me what the authors mean by "small Sb-displacements could not be resolved by single crystal XRD since XRD probes the averaged static electron density". On one hand, SC-XRD as a bulk probe is very sensitive to small distortions averaged over the crystal. If, on the other hand, the authors refer to the varying distortion on the atomic scale because of disorder, their conclusion is correct. However, the DFT calculations they present in the following are not able to capture these locally varying distortions because of disorder either because DFT of a bulk unit cell uses the same average static density as SC-XRD. The authors should clarify these discrepancies.

13) Regarding ARPES: Why did the authors only integrate in the range $[-50,-70\text{meV}]$ to obtain a Fermi surface in Fig. 4a and not $[0,-70\text{meV}]$? Technically the range used in the paper is not a Fermi surface.

14) Regarding ARPES: Could the authors please provide the polarization of the incident light?

15) Regarding DFT: Panels Fig 4c and similarly Figs. 6b,c are presented in a confusing way. The figure captions describes these as Fermi surfaces but what seems to be shown is a heatmap-style plot of $E(k_x,k_y)$ for the valence band and only the dashed lines correspond to actual constant energy contours, although none of those are shown for the $E=EF$. We suggest the author plot actual Fermi surfaces in all cases to avoid confusion.

16) Regarding DFT: Was spin-orbit coupling included in the calculations?

17) Regarding DFT: Could the author explain in more detail why "the band dispersion measured along [Fig. 4(b)] corroborates the presence of the Dirac cone"? Even though all DFT calculations presented in this paper show step and linearly dispersing bands near the X point but none of these calculations actually show a band crossing (not even above EF) that would qualify those linear dispersing bands as Dirac bands.

18) Regarding electrical transport: Since the author discuss the Hall data for almost one full paragraph it might make sense to show them in the main text.

19) Regarding electrical transport: Can the authors please discuss the Zeeman splitting of the quantum oscillations in more detail? The discussion does not lead towards that feature and it appears quite unexpected in the text.

20) Regarding electrical transport: The use of the intercept of the Landau level fan diagram as a proof for 3D rather than 2D nature of the Fermi surface is not very convincing. Are there other possible explanations for the deviation of the intercept from 0.5?

► Reply to the comment of Reviewer #1

General comments

The authors present a comprehensive study on a high-entropy alloy of $A^5\text{MnSb}_2$. They are able to successfully grow the material. By various structural microscopies, the crystal structure is determined. ARPES measurements are further carried out and the band structure, consisting of a hole pocket at the Γ point and a linear dispersion around the X point, is mapped out. The band structure is confirmed by DFT calculations. Quantum oscillations reveal a nontrivial Berry phase of π , suggesting a Dirac band. The field angle dependence of the oscillation frequency indicates a 3D Fermi surface. The experiments were carefully carried out and data are of high quality. The idea of high-entropy engineering is novel. The fact that the carrier mobility remains high in spite of disorder is surprising and welcome. Overall, the manuscript is convincing. My biggest concern is about identification of the Dirac band by electrical transport.

Response: We would like to thank the reviewer for reading our manuscript carefully and for the positive assessment of our work. Our response to your concern about the transport measurement is given below.

Technical comment 1

Both ARPES and DFT calculations indicate that the hole pocket at the Γ point is much larger than the Dirac pocket at the X point. It is natural to believe that this band will dominate the transport, unless the mobility of this band is extremely low. It would be nice to see an estimation of the upper limit of the mobility, based on the relative sizes of two pockets. It is stated on page 5 that temperature-independent linear ρ_{xy} - B curves indicate that the transport of this crystal is dominated by the linear Dirac bands. Please elaborate the reasoning behind this statement.

Response: Following the reviewer's suggestion, we have estimated the mobility ratio between those from Fermi pockets at Γ point (μ^Γ) and X point (μ^X), μ^Γ/μ^X , as reported below. According to the standard Drude formalism, mobility can be expressed in terms of electron charge (e), mean-free time (τ), and effective mass (m^*):

$$\mu = \frac{e\tau}{m^*}. \quad (\text{R1})$$

In addition, by using the relation of m^* to Fermi velocity (v_F) and Fermi vector (k_F), i.e., $m^* = \frac{\hbar k_F}{v_F}$, one can obtain the following equation,

$$\mu = \frac{e\tau}{\hbar} \frac{v_F}{k_F}, \quad (\text{R2})$$

where \hbar is reduced Planck constant. Assuming that the mean-free time is the same for both the bands at Γ and X points, the relative mobility ratio can be written as

$$\frac{\mu^\Gamma}{\mu^X} = \frac{v_F^\Gamma/v_F^X}{k_F^\Gamma/k_F^X}. \quad (\text{R3})$$

FIG. R1: ARPES band mapping along the Γ -X direction (identical to Fig. 4(b) of the main manuscript) with the solid/dashed lines serving as the eye guide to show the pocket sizes and band slopes at Γ and X points.

The ARPES data shown in Fig. R1 allows us to estimate v_F^Γ/v_F^X and k_F^Γ/k_F^X ratios based on the slope of the band dispersions and pocket sizes, respectively:

$$v_F^\Gamma/v_F^X = 7.09 \times 10^{-2},$$

and

$$k_F^\Gamma/k_F^X = 3.84.$$

Finally, μ^Γ/μ^X is 1.8%, showing that the mobility of the band at Γ point is two orders of magnitude smaller than that at X point.

Additionally, we note that the band at the Γ point has some contributions from the orbitals of the A cation, as seen from the atom-resolved band structures calculated for BaMnSb_2 , SrMnSb_2 , and CaMnSb_2 (Fig. R2). This implies that in the high-entropy crystal, the transport originating from the Γ band is disturbed by the randomness at the A site. Therefore, we expect that the τ^Γ is much smaller than τ^X , indicating that the actual mobility ratio is smaller than the value estimated above.

The reason behind the statement “temperature-independent linear ρ_{xy} - B curves indicate that the transport of this crystal is dominated by the linear Dirac bands” is the following. Within an approximation of a single-band model, ρ_{xy} depends linearly on the magnetic field B , whereas it deviates from linearity in the case of a two-band model. Thus, a linear ρ_{xy} - B curve obtained for the high-entropy crystal suggests that either the hole pocket at the Γ or X point dominates the transport, rather than both of them contributing to the transport. Given the very low mobility of the band at the Γ point compared with that at the X point as shown above, one can conclude that the transport behavior of this compound is dominated by the band at the X point, i.e., the Dirac band. In addition, the slope of the ρ_{xy} - B , in principle, depends on the carrier density and thus temperature. However, the change in carrier density is expected to be small within a narrow temperature window (such as 2–50 K used for our measurements), and the ρ_{xy} - B curve becomes almost temperature

FIG. R2: Atom-resolved band structure calculated for (a) BaMnSb₂, (b) SrMnSb₂, and (c) CaMnSb₂.

independent as shown in revised Fig. 5(a). Such behavior has also been confirmed in the pristine BaMnSb₂, for which the X-point hole pocket is proved to dictate the transport property via the observation and analysis of the quantum Hall effect [*Nat. Commun.*, **12**, 4062 (2021)]. Overall, we have concluded that linear ρ_{xy} - B curves are indicative of the single-band (those at X point in this case) dominated transport of this crystal. We have added these discussions as Supplementary Note 1 to Supplementary Information.

Technical comment 2

Determination of the Berry phase by quantum oscillations can be tricky and susceptible to errors, when multiple frequencies or Zeeman splitting are involved, or the low Landau level indices are not available. The more Landau level indices in the Landau fan diagram, the more reliable the fitted intercept would be. In Fig. 5b, at least a peak and a valley for $1/B > 0.06$ are well resolved, but these valuable data points are not included in the Landau plot. Please include them in fitting. Moreover, can the phase analysis be performed for oscillations at high field angles where the oscillations are clean, as seen in Fig. S4b? Such analysis would certainly strengthen the arguments.

Response: According to this comment, we performed Landau-level (LL) fan diagram analysis after adding the points of $1/B > 0.06$ for $\theta = 0^\circ$ data as well as for high field-angle transport data ($\theta = 84^\circ$). Figure R3(a) shows the LL fan diagram obtained from ρ_{xx} data at 50 K ($\theta = 0^\circ$), where a peak and valley point with $1/B > 0.06$ were included in addition to those used in the original plot, Fig. 5(c). The linear fitting to the new diagram and the extrapolation yields the intercept n_0 of 0.75(7); this value matches well with the value presented in our original manuscript [0.67(14)] within error and is consistent with the expectation value for materials with a 3D Fermi surface [0.625 or 0.375].

FIG. R3: Landau fan diagram obtained from ρ_{xx} data (a) at 50 K ($\theta = 0^\circ$) and (b) at 1.7 K ($\theta = 84^\circ$).

Figure R3(b) displays the LL fan diagram constructed from the high field-angle data at 1.7 K. The fitting and extrapolation to this diagram result in an intercept of 0.62(9), which is also very close to the expectation value. Based on these additional analyses, we replaced the original Fig. 5(c) with the new diagram [Fig. R3(a)] and added Fig. R3(b) into Supplementary Information. We thank the reviewer for making this comment, which has been extremely helpful in strengthening our arguments.

Technical comment 3

How does the oscillation frequency compare to the pocket size seen in the ARPES? An agreement between two may provide a consistency check for the Dirac band.

Response: We are grateful to the reviewer for the suggestion. Although the estimation of the pocket size from the ARPES data contains a large error due to the limited resolution, we estimated the cross-sectional area from Fig. R1, where the estimated size of the pocket along k_x direction was 0.096 \AA^{-1} . This value leads to a cross-sectional area in 0.72 nm^{-2} in k_x-k_y plane when assuming a circular shape. The observed oscillation frequency of 79 T corresponds to the cross-sectional area of 0.75 nm^{-2} based on Onsager's relation, which is very consistent with the extracted value from the ARPES data. We anticipate that this agreement between quantum oscillation and ARPES provides a consistency check. We have added this information to the revised manuscript.

Technical comment 4

A cyclotron mass is obtained from the temperature dependence of the oscillation amplitude. It is rather large, $0.5m_0$. The authors argue that the Dirac node may be massive (By the way, a reference needs to be provided in the manuscript). But, the dispersion of this band seems quite linear at the Fermi level according to the ARPES data. How can a steep linear dispersion generate such a large cyclotron mass? Is it possible that the oscillation is in fact from the large trivial hole pocket?

Response: We thank the reviewer for this valuable comment. First, we would like to further explain the characteristics of the Dirac nodes identified for the $AMnX_2$ family ($A = \text{Ba, Sr, Ca, Eu, and Yb}$; $X = \text{Sb and Bi}$). The crystal structure (which depends on the size of A cation) and the magnitude of spin-orbital coupling (i.e., the chemistry of the X anion) dictate the band dispersion around the X point. The structures without X -displacement ($P4/nmm$ and $I4/mmm$ structures, Fig. 1 of the main manuscript) exhibit massless Dirac crossing at the X point with SOC excluded. On the other hand, activating SOC or X -displacement (zig-zag chain formation in X_4 layers) leads to a small gap at the Dirac point. Indeed, large electron masses have been observed for YbMnSb_2 [$0.134m_0$, *Phys. Rev. Materials* **2**, 021201(R)] and SrMnBi_2 [$0.29m_0$, *Phys. Rev. Lett.* **107**, 126402]. Since the high-entropy compound also involves Sb-displacements as visualized by STEM analysis, we anticipated massive Dirac fermion behavior for this material as well.

The difference in the chemical potential between the bulk and surface, which is proved by transport and ARPES, respectively, would be the possible reason of the large cyclotron mass despite the linear dispersion seen in ARPES. Since the Dirac node is gapped, a slight deviation in chemical potential toward a less hole-doped regime pushes the Fermi energy to a nonlinear $E-k$ dispersion region, giving rise to a massive nature of the electron. We have added these discussions to Supplementary Note 2.

► Reply to the comments of Reviewer #2

General comments

This study introduces an innovative approach to create materials with relativistic electronic states, especially Dirac/Weyl semimetals, which are pivotal in current condensed matter physics and materials science. It demonstrates that by harnessing high configuration entropy, the $AMnSb_2$ Dirac material family can be modified to yield a unique polar structure with exceptional properties absent in the original compounds. This process results in the development of an entropy-stabilized high-entropy Dirac material, maintaining linear band dispersion despite lattice irregularities. Additionally, the study uncovers a remarkable transformation from two-dimensional Dirac states to highly anisotropic three-dimensional Dirac states in the high-entropy phase, substantiated by experimental data and theoretical models. As such, I would like to recommend accepting the article for publication after the authors address the following comments:

Response: We would like to thank the reviewer for taking the time to review our manuscript and identifying the novelty of our work. We appreciate his/her recommendation and questions, to which our response is listed below.

Technical comment 1

Phonon behaviors of the $AMnSb_2$ series are not discussed in the manuscript. Could the authors provide comments on or perform theoretical calculations to enhance the discussion?

Response: Thank you so much for the comments. First, we would like to note that phonon calculations for this high-entropy crystal are difficult due to the following reasons.

- Modeling the random distribution of the *A*-site cations requires prohibitive computational resources.
- Even if single-cation approximation is employed for the *A* site, the fixed lattice parameters to the experimental values hinder full structural relaxation (atomic positions and lattice parameters), and the subsequent phonon calculation will lead to a bunch of imaginary phonon modes, yielding rather meaningless results.

Instead, we measured the Raman spectrum to address the phonon behavior of the high-entropy crystal and compared it with those collected for parent compounds, $BaMnSb_2$, having the same point group $mm2$ as the high-entropy compound. Figure R4(a) shows the Raman spectra at room temperature, where one can find that a peak of the high-entropy crystal is slightly shifted toward the low wavenumber region when compared to the $BaMnSb_2$. The ticks at the bottom of Fig. R4(a) indicate the DFT-calculated Raman peak positions for $BaMnSb_2$. The phonon frequency may be underestimated because of the harmonic approximation and 0-K nature of DFT. Suppose we assign the strongest Raman peak to the theoretical peak position at around 135 cm^{-1} , the peak corresponds to an A_1 mode, for which eigendisplacement pattern is shown in Fig. R4(b). Sb displacements in

FIG. R4: (a) Raman spectrum collected for $A^5\text{MnSb}_2$ and pristine BaMnSb_2 . (b) Eigendisplacement of the A_1 mode of BaMnSb_2 ($\approx 135 \text{ cm}^{-1}$).

the Sb_4 layer is the main component of this phonon mode. Therefore, the peak shift toward the low wavenumber region observed for the high-entropy crystal implies the softening of Sb-displacement motion, probably due to the enlarged cavity caused by smaller cation accommodation than Ba ions.

We have added a comment on the phonon property of the high-entropy compounds in the discussion section and included the above discussion as well as the figures in the Supplementary Information (Supplementary Note 3).

Technical comment 2

Ideally, achieving a homogeneous solid solution is challenging in high-entropy materials. Can the authors comment on whether they observed any form of short-range order in their experiment?

Response: We agree with the comment and thank the reviewer for bringing up this point. We think that short-range ordering manifests itself in diffuse reflection spots in diffraction patterns. In our selected-area electron diffraction experiment, however, we did not observe any diffuse reflections except a very small streak mentioned in our response to comment 2 from Reviewer #3. We have added a sentence to the main manuscript (page 2, right column) to clarify that short-range ordering is less likely to exist as far as we are aware.

Technical comment 3

High entropy is associated with intrinsic structural disorder. How does this type of disorder affect the transport properties of the Dirac material?

Response: In the case of $A^5\text{MnSb}_2$ having a layered nature, the layers which accommodate

high-configurational entropy and Dirac electrons are different; the former is the $A\text{-MnSb}_4$ slab, and the latter is the Sb_4 zig-zag chain layer. Therefore, the disorders in the $A\text{-MnSb}_4$ slab have limited impact on the mobility of Dirac Fermions on the Sb_4 layers. As discussed by Pixley *et. al.* [*Phys. Rev. Lett.* **115**, 076601], Dirac state is robust against a certain magnitude of disorder. Thus, we believe that the transport of Dirac materials is not severely affected by disorder up to the critical point; our $A^5\text{MnSb}_2$ case falls into this regime due to the layered nature as mentioned above. Beyond the critical point, however, diffuse metal and an insulating state appear, which may be realized when the high-configurational entropy is directly introduced in the atomic layer hosting Dirac electrons. The above discussion is added in Discussion section of the revised manuscript.

► Reply to the comments of Reviewer #3

General comments

In this manuscript, Laha and coauthors report the single crystal growth and characterization of a ternary antimonide $AMnSb_2$, where high configurational entropy was achieved on the cationic A site by mixing five different cations. Using a combination of local and bulk probes they identify a new, non-centrosymmetric crystal structure that is not found in any of the parent, single cation $AMnSb_2$ materials. Based on angle-resolved photoemission spectroscopy and quantum oscillation data they show that the high entropy $AMnSb_2$ maintains a well-defined Fermi surface dominated by linear Dirac bands despite the high degree of disorder on the A site. Moreover, they claim that the dimensionality of the Fermi surface is found to change to 3D as compared to 2D in the parent, single cation $AMnSb_2$.

The work that Laha and coauthors present is conceptually well designed, and the suite of various techniques used in this work is novel for high entropy materials. In particular, the comparison of the high entropy $AMnSb_2$ with the permutations of binary solid solution $(A,A')MnSb_2$ is particularly thorough and shows that entropy is the driving force for stabilizing the new crystal structure observed for the high entropy $AMnSb_2$. Furthermore, the observation of quantum oscillations up to a temperature of 50 K in such a disordered material is remarkable and shows the high quality of the crystals.

However, we have three significant concerns with the present work. First, the evidence in support of the main claim of realizing new structures by high entropy alloying, the determination of the crystal structure and in particular the breaking of inversion symmetry is not convincing. Second, the design criteria applied here and the general applicability of the high entropy alloying approach to generate new structures is not discussed well in the paper. Third, the claimed distortions to the Sb chains in this material are in direct contradiction with the observed quantum oscillations and moreover the evidence for the change in dimensionality is not convincing. For these reasons, we cannot recommend this paper for publication in Nature Communications in its current form. If properly addressed, these aspects would increase the impact of the paper considerably.

If the authors could please reply to the following comments.

Response: We would like to thank the reviewer for the careful review and insightful comments, and for finding our work well-designed. Our responses to the reviewer's comments are given below, and we have revised the manuscript accordingly.

FIG. R5: SHG intensity as a function of the polarization angle of the incident light collected for (a) $A^5\text{MnSb}_2$ and (b) YbMnSb_2 .

Technical comment 1

The observation of “a significant signal” in second harmonic generation (SHG) is taken as key evidence for inversion symmetry breaking without providing details of what a significant signal is. In particular, the authors do not describe how they disentangle bulk and surface contributions to the signal, which is crucial as surfaces also break inversion symmetry and SHG is surface sensitive. We suggest the authors improve the discussion of SHG data and in addition provide measurements on a reference sample with inversion symmetry. Measuring one of the centrosymmetric, single cation AMnSb_2 materials for example lends itself to a straightforward comparison with the high entropy AMnSb_2 .

Response: Following the reviewer’s suggestion, we have measured SHG with the same optical setup for YbMnSb_2 , a pristine compound with inversion symmetry, and the result is depicted in Fig. R5(b) together with the data collected for the high-entropy crystal [panel (a), which is identical to Fig. 3(a) in the main text]. Please note that the scale of the radial axis of panel (b) is five times smaller than that of panel (a). Given the centrosymmetric nature, the surface-originated SHG is the main contribution to the signal from YbMnSb_2 , but the intensity is within the background level. Because of the similarity of the structural motif between the two compounds as well as the fact that both the signals were collected from (001) plane, we expect that the surface contribution to the SHG signal in the high-entropy crystal is of the same order of magnitude. By comparing panels (a) and (b) of Fig. R5, one can find that the signal from the high-entropy crystals is much stronger than that from YbMnSb_2 despite the comparable surface contributions. Therefore, we concluded that the SHG signal observed in the high-entropy crystal originates from the intrinsic effect due to the lifted inversion symmetry in the crystal structure with an almost negligible contribution from the surface.

In the revised manuscript, we have mentioned that the SHG signal from the high-entropy crystal was much stronger than that from centrosymmetric YbMnSb_2 , and Fig. R5 has been included in Supplementary Information.

FIG. R6: (a) ADF-STEM image along [010] zone axis, (b) fast Fourier-transformed image of panel (a), and (c) enlarged view of SAED pattern [panel (d) of Fig. 3].

Technical comment 2

Based on the observation of $(00l)$ $l = \text{odd}$ reflections in the SAED pattern, which are forbidden in the centrosymmetric space groups of the single cation $AMnSb_2$ materials, the authors perform group theoretical analysis to derive the polar space group of the high entropy $AMnSb_2$. However, it is known from related materials with the same space group ($Pnma$, #62), that stacking faults can give rise to these otherwise forbidden reflections, e.g. in *Phys. Rev. B* 54, 13587 (1996). What efforts have the authors taken to exclude stacking faults or other reasons as the origin of the $(00l)$ $l = \text{odd}$ reflections?

Response: We thank the reviewer for bringing this reference to our attention. We acknowledge that it is very difficult to exclude the possibility of stacking faults and sample bending. The reference studied $LaTeSb$ with the space group of $Pm\bar{c}n$ (a non-standard setting of $Pnma$), where the stacking of the Sb sheets is along the c axis. As pointed out by the reviewer, the authors of the reference found forbidden $hk0$ ($h + k$: odd) reflections in the SAED patterns along $[001]$ zone axis and attributed it to the stacking fault and/or sample bending. In our SAED experiment, $00l$ (l : odd) reflections were observed in the SAED pattern along the $[010]$ zone axis. Therefore, we would like to note that our situation is somewhat different from this reference.

We understand that it is important to make efforts to exclude the possibility of stacking fault and bending. Figures R6(a) and R6(b) show the STEM image along $[010]$ zone axis and its fast Fourier-transformed image, respectively. As seen in Fig. R6(b), no streak line is observed, which suggests the absence of a stacking fault in this region. On the other hand, when the SAED pattern [Fig. 3(d)] was thoroughly investigated, we did observe very weak streaks around the 001 reflection spot [Fig. R6(c)]. Overall, we cannot completely eliminate the possibility of stacking faults in this material. However, the streak is not continuous along c^* direction, and thus we expect that the stacking fault is, if it exists, not so influential to allow otherwise forbidden reflections. In the revised manuscript, the possibility of stacking fault is discussed in Supplementary Note 5, which is now mentioned in the revised main manuscript.

Technical comment 3

In the structure determination, the authors employ a range of techniques, which is commendable. However, they are also conflating local and average structure. It is well known in high entropy materials that very high levels of local distortions, due to their random nature, are not necessarily representative of the average crystal structure. In terms of establishing the true average space group for this material, we are not convinced by the authors claims that single crystal x-ray diffraction is an unsuitable method here (or even, for that matter, powder x-ray diffraction). At a minimum, the authors should present goodness of fit parameters across all the candidate space groups (including the centrosymmetric options) and show that they are not distinguishable on that basis.

Response: We are thankful for the comment, and we apologize for the confusing description of x-ray diffraction. We agree with the reviewer that the x-ray diffraction is able to well capture the average structure and symmetry even though it is high-entropy materials and have revised the sentence to avoid confusion. Following the reviewer's suggestion, we here present the results and statistics about the structural refinement of the single-crystal XRD data. We identified a total of eight candidate space groups: $Pcmn$ (identical to that of pristine SrMnSb_2) $P2_1/c$, $P2_1/n$, $P2_1/m$, $P2_12_12_1$, $P2_1mn$, $Pcm2_1$, and $Pc2_1n$, where non-standard settings are used for some of them so that the c axes indicate the stacking direction. The former four space groups are centrosymmetric, and the others are noncentrosymmetric. We have performed structural refinements with all these candidates, but due to excessively high R values, the refinements based on centrosymmetric space groups were deemed unfeasible. In contrast, the refinement was successfully performed using the four noncentrosymmetric space groups. The statistics and crystallographic information on the refined structures are now included in the revised Supplementary Information as Tables S1–S5. Reflecting comment No. 11 from Reviewer #3, a single occupation of Ba is assumed for the A site.

The goodness of fit indicators for the four space groups are very similar: 1.072 for $P2_12_12_1$, 1.132 for $P2_1mn$, 1.095 for $Pmc2_1$, and 1.184 for $Pc2_1n$. Thus, we cannot unambiguously determine the structure only in terms of the fitting factors of structural analysis. However, as listed in Table. I of our main manuscript, only the $P2_1mn$ can be consistent with our observation of $00l$ (l : odd) reflections among the four candidate noncentrosymmetric space groups. Overall, we have concluded that the high-entropy $A^5\text{MnSb}_2$ crystal adopts $P2_1mn$ structure.

The goodness of fit indicators for the four noncentrosymmetric candidates have been included in the revised manuscript. In addition, as pointed out by comment No. 10 from Reviewer #3, we have added tables containing statistic and crystallographic information in Supplementary Information as Tables S1–S5.

Technical comment 4

Related to the previous questions, can the authors describe how representative for the whole sample that SAED pattern is? In the methods section, it is stated the one lamella was used for STEM but is there any variation within that lamella or between different lamellae?

Response: Although one lamella was extracted for STEM study as stated in the methods section,

the lamella is across the 90° structural domain wall and allows us to study two different domains (in other words, two different orientations). The SAED patterns were acquired using a circular aperture that spans about 800-nm projected diameter. Multiple SAED patterns were obtained from different regions, and the results are consistent with each other. To clarify this point, we have modified the methods section.

Technical comment 5

The introduction suggests that the high entropy alloying approach can be used to overcome limitations of traditional (binary) solid solution approaches and create new crystal structures. However, it is unclear to us whether this approach is generally applicable or if it is coincidence that it works in $AMnSb_2$. To this end, it would help if the authors could clarify the general applicability of the approach and allude to the underlying mechanism of creating new crystal structure by high entropy alloying, if it exists.

Response: We thank the reviewer for the comment. We anticipate that the high-entropy strategy to create a new structure inaccessible by the binary solid-solution approach is generally applicable. The mechanism behind the emergence of a new crystal structure is a *compromise* among crystalline phases with distinct structures (or symmetries). In our case, there are three different polymorphs adopted by the $AMnSb_2$ series ($P4/nmm$, $Pcmn$, and $I2mm$); two of them prefer aligned $MnSb_4$ layers with centrosymmetry, while the other crystallizes in a noncentrosymmetric structure with anti-aligned $MnSb_4$ layers. The resulting high-entropy structure possesses aligned $MnSb_4$ layers and a noncentrosymmetric structure, which can be viewed as a compromising consequence of the distinct structural features.

In addition to our example, there are a few cases where high entropy gives rise to different crystal or magnetic structures absent from the pristine systems. The former case is $(Ti,Zr,Hf,Sn)O_2$ [*Communications Materials* **4**, 45 (2023)]. Although $(Ti/Sn)O_2$ and $(Zr/Hf)O_2$ crystallize in rutile and baddeleyite structures, respectively, the high-entropy phase adopts an α - PbO_2 structure. The α - PbO_2 structure is a higher-symmetry polymorph of baddeleyite, but the local coordination environment around the cation site resembles rutile (coordination number = 6) rather than that of baddeleyite (coordination number = 7). Again, the high-entropy structure can be regarded as a compromising result of distinct structures. A similar compromising phase appears in the case of magnetic structure, as exemplified by $(Gd,Tb,Dy,Ho)Mn_6Sn_6$ [*Communications Physics* **5**, 63 (2022)]. Gd and Dy favor the in-plane and out-of-plane spin ordering, respectively, whereas both Dy and Ho prefer tilted spin orientation (45° and 50°, respectively). As a result, the magnetic ground state of the high-entropy crystal has a 30° tilted spin orientation, which does not exist in any of the parent compounds.

Given that different structures whose chemical compositions are akin to each other are identified, one can mix them to achieve a new structure by utilizing this compromise mechanism. In this sense, we believe that this strategy is widely applicable. Of course, such a compromise also plays a crucial role in the cases of binary solid solutions; however, an insufficient configurational entropy often ends up with a phase separation as observed in $(Ba,Ca)MnSb_2$ and $(Ba,Yb)MnSb_2$ [Figs. 2(b) and 2(c) of the main manuscript]. Thus, high configurational entropy needs to be employed to

merge distinct structures into a homogeneous new structural phase.

To clarify these points, we have added paragraphs in the Discussion section of the main manuscript and Supplementary Information.

Technical comment 6

Related to the previous question, the design principle employed here is not explained. High entropy materials are most frequently taken to be equimolar. Can the authors explain why they chose the composition with 40% Ba over for example equimolar concentration (20%) for all cations? The actual magnitude of configurational entropy is significantly reduced from the equimolar limit.

Response: As pointed out in the original manuscript, our target composition is $(\text{Ba}_{0.4}\text{Sr}_{0.15}\text{Ca}_{0.15}\text{Eu}_{0.15}\text{Yb}_{0.15})\text{MnSb}_2$ and is different from the equimolar composition $(\text{Ba}_{0.2}\text{Sr}_{0.2}\text{Ca}_{0.2}\text{Eu}_{0.2}\text{Yb}_{0.2})\text{MnSb}_2$. The reason why we chose this target composition is the following. First, we would like to emphasize that our composition still has a large configurational entropy ($\approx 1.50R$; R is the universal gas constant), close to the equimolar ideal case with five different cations ($\approx 1.61R$) and higher than the four-component solid solutions with equimolar ratio ($\approx 1.38R$).

Second, the reason behind the Ba-40% composition is that the crystal structure of the AMnX_2 series ($A = \text{Ba}, \text{Sr}, \text{Ca}, \text{Eu}, \text{and Yb}$; $X = \text{Sb and Bi}$) is sensitive to the atomic size of A cations as depicted in Fig. 1 of the main manuscript. Among the pristine compositions, only the Ba-containing compounds (BaMnSb_2 and BaMnBi_2) show polar structure ($I2mm$), suggesting that large A cation plays an important role in stabilizing the polar symmetry. Building on this implication, we chose the Ba-rich composition aiming to create a high-entropy polar phase. To clarify these points, some sentences are added in the revised manuscript.

Technical comment 7

Regarding the 2D-3D transition: Can the authors please give a more detailed explanation for the change in Fermi surface dimensionality upon high entropy alloying? We are not convinced that the small “rumpling” of the Sb layer gives rise to a significant out of plane dispersion. Naively, one would assume that the interlayer coupling between neighboring Sb layers has to increase significantly in order to obtain a notable out of plane dispersion. However, the rumpling assumed in the calculations seems to be rather small, so I do not understand how this would create the 3D Fermi surface. In addition, Fig. 6c in the way it is presented shows that the (presumably) valence band is still highly 2D with only a very small 3D component. Furthermore, such “rumpling”, random displacements in the Sb chains, would seem to be in direct contradiction with the observed quantum oscillations. Photon energy dependent ARPES data would be much stronger evidence of 3D vs 2D character – have the authors performed this measurement?

Response: We agree with the reviewer’s comment that the interlayer coupling between neighboring Sb layers has to increase significantly to obtain a notable out-of-plane dispersion. Since

the high-entropy A-cation layers are also adjacent to MnSb_4 tetrahedral layers, we expect such out-of-plane displacive distortion to also exist in the MnSb_4 layers (specifically, Sb layers and Mn layers). The out-of-plane displacement within MnSb_4 layer would assist the electron tunneling across the adjacent Sb_4 layers, increasing the coupling between them. We anticipate that the indirect distortion induced in the MnSb_4 layers also contributes to the formation of a quasi-3D Fermi surface, which is now mentioned in the revised manuscript.

Regarding Fig. 6(c), the reviewer is correct. The Fermi morphology is still highly 2D-like, and the out-of-plane component is very small compared with the in-plane one. Therefore, we mentioned that the Fermi surface is highly-anisotropic (i.e., quasi-3D) instead of strong 2D in the revised manuscript. To avoid confusion, we have replaced “3D” with “quasi-3D” in the revised manuscript.

We understand what the reviewer pointed out, i.e., the apparent inconsistency between the randomness in the Sb_4 layers and the observation of quantum oscillations. As discussed in *Phys. Rev. Lett.* **115**, 076601, however, Dirac state is robust against the disorder up to a certain quantum critical point. Therefore, one can still expect the observation of quantum oscillation even with the random rumpling distortion. Such a robustness is mentioned in Discussion section of the revised manuscript.

As for energy-dependent ARPES, we did not conduct such an experiment primarily because of the difficulty in getting a large cleaved surface area and the limited size of the cleaved surface. The attempt to vary photon energy by rotating the monochromator presents a technical challenge, as adjusting the photon energy in this manner would cause the light spot on the sample to move away from the clean surface. Hence, performing photon energy-dependent ARPES experiments poses a significant technical hurdle in our sample.

Technical comment 8

Regarding the abstract: The exact composition of the material should be provided in the abstract. As written, it is implied that the material is equimolar, which it is not.

Response: Thank you so much for your suggestion. We have modified the abstract to include the exact composition of the material.

Technical comment 9

Regarding the STEM data: the authors claim that SAED has “superior detection sensitivity to conventional x-ray based techniques”. Can they elaborate more on what feature refer to in terms of superior detection sensitivity and why SAED is better than X-ray. Including a reference would be helpful here.

Response: We apologize for the incorrect statement. This may be the case when comparing SAED and powder XRD, especially if the sample contains a heavy element with strong x-ray absorption, due to which powder XRD peaks get weaker and will be buried under the background. The sentence has been corrected in the revised manuscript.

Technical comment 10

Regarding SC-XRD: For single crystal XRD data it is common practice to include the results table, including space group, number of reflections, R values and goodness of fit for the refinements etc. Could the authors please provide these data in the supplementary material?

Response: Following the reviewer's suggestion, we have added the tables containing such information in the Supplementary Information (Tables S1–S5).

Technical comment 11

Regarding SC-XRD: Based on what argument is Eu used as the cation in the refinement? Based on the composition from EDS the average electron count on the A site should be ≈ 51 (much closer to Ba than to Eu), which would result in an occupation of the A site of only 0.8. Again, these results should be reported in the supplementary. In general, the choice of cations for various modellings (XRD, DFT) appears to be rather inconsistent. Could the author please explain why they chose certain cations for certain models?

Response: We thank the reviewer for this valuable comment. Instead of refining the occupancy parameter for each of the five cations sitting on the A site, we modeled the electron density of the high-entropy A -site by varying the occupation of a single A cation. If Ca or Sr—which has fewer electrons than the average electron count—is chosen as the A cation, the occupation will exceed 1, which is physically meaningless. We did carry out the refinement with assuming Ca for A site, but the refinement was not stable. If we choose Ba—which has a close number of electrons to the average—as the single cation, there is still a possibility that the refined occupancy parameter surpasses 1. Therefore, we chose Eu carrying more electrons than Ba to model the A -site electron density and refine its occupancy. As the reviewer indicated, the refined occupancy for the A site (Eu) is close to 80%. At the same time, we agree with the reviewer that it would be more natural to use Ba for the A site based on the electron count. We also performed the refinements using Ba, and the final occupancy for the A site (Ba) was about 90% as listed in Tables S1–S5. Reflecting the lower electron count of Ba than Eu, the refined occupancy is larger for Ba. Based on this comment, we listed the results with Ba in the Supplementary Information.

As for the DFT modeling, we calculated the band structure by assuming 100%-occupation of each of Ba, Sr, or Ca for the A site, and the results are depicted in Figs. 4(d), S10(b), and S10(c). Here, we exclude Eu and Yb to avoid the complexity of the band structure due to the contribution from f orbitals. In the case of Ba and Sr, the $P2_1mn$ phase was converted to $Pcmn$ when relaxing the atomic positions, whereas it remained in the $P2_1mn$ symmetry when Ca resides at the A site. In addition, this relaxed structure has the closest Sb-displacement values to those observed in the STEM image [Fig. S10(a)]. Therefore, we chose Ca for the A site of our DFT model. Please note that as seen from Figs. 4(d), S10(b), and S10(c), the band dispersion around the gapped Dirac node (X point) is robust against the choice of alkaline-earth A cation. We have modified the main manuscript to explain the reasons of the cation choices.

Technical comment 12

Regarding SC-XRD: It is not clear to me what the authors mean by “small Sb-displacements could not be resolved by single crystal XRD since XRD probes the averaged static electron density”. On one hand, SC-XRD as a bulk probe is very sensitive to small distortions averaged over the crystal. If, on the other hand, the authors refer to the varying distortion on the atomic scale because of disorder, their conclusion is correct. However, the DFT calculations they present in the following are not able to capture these locally varying distortions because of disorder either because DFT of a bulk unit cell uses the same average static density as SC-XRD. The authors should clarify these discrepancies.

Response: We apologize for the inaccurate sentence and potential confusion. As long as the displacement is static, it can be detected by single crystal XRD. Indeed, the Sb displacements were partially captured by our revised refinement against the single-crystal XRD pattern. Based on the crystal structure listed in revised Table S3, the two Sb–Sb bond lengths in the Sb₄ layers are ≈ 2.147 and ≈ 2.234 Å. However, these values are still somewhat different from those detected by STEM observation, 2.267(17) and 2.402(7) Å. To include the static Sb displacement properly, we utilized structural relaxation while fixing the lattice parameters to the XRD-obtained values. From this structural relaxation we found that the structure with $A = \text{Ca}$ has the Sb–Sb bond lengths [Fig. S10(a)] closest to what is observed in the STEM image. This is why we used $A = \text{Ca}$ structural model in the following band structure calculations. Overall, we meant displacements averaged over the crystal, which is generally considered in single-crystal XRD and DFT calculations. Following this discussion, we have changed some sentences in the revised manuscript.

Technical comment 13

Regarding ARPES: Why did the authors only integrate in the range $[-50, -70 \text{ meV}]$ to obtain a Fermi surface in Fig. 4a and not $[0, -70 \text{ meV}]$? Technically the range used in the paper is not a Fermi surface.

Response: The reviewer is correct, and the labeling of the integration map with an energy range of $[-50, -70 \text{ meV}]$ as a Fermi surface is inaccurate. In the revised version, we have corrected it and labeled it as a constant energy contour.

Technical comment 14

Regarding ARPES: Could the authors please provide the polarization of the incident light?

Response: In our ARPES experiment, the linear horizontal polarization of incident light is used. This point has been clarified in the methods section of our revised manuscript.

Technical comment 15

Regarding DFT: Panels Fig. 4c and similarly Figs. 6b,c are presented in a confusing way. The figure captions describes these as Fermi surfaces but what seems to be shown is a heatmap-style plot of $E(k_x, k_y)$ for the valence band and only the dashed lines correspond to actual constant energy contours, although none of those are shown for the $E = E_F$. We suggest the author plot actual Fermi surfaces in all cases to avoid confusion.

Response: We thank the reviewer for this comment apologize for the confusion. The reason why we plotted in such a way is to represent the hole-doped nature of the high-entropy crystal, which is confirmed via ARPES and Hall measurements. We have clarified this point in the revised manuscript.

Technical comment 16

Regarding DFT: Was spin-orbit coupling included in the calculations?

Response: Spin-orbital coupling (SOC) was not included in the calculations shown in the main manuscript. The inclusion of SOC further opens the gap at around X point, which is already opened by Sb displacements even without SOC being included. Other impact originating from SOC on the band structure is marginal in the $AMnSb_2$ series; for example, one can find the comparison in the case of pristine $CaMnSb_2$ in *Phys. Rev. B* **103**, 245104 (2021). We anticipate that the simplification by excluding SOC has no impact on our conclusion.

The inclusion of SOC requires us to input the precise magnetic configuration of the Mn sublattice. Still, it remains unknown for the high-entropy crystal and is beyond the scope of this study. Given that $SrMnSb_2$, $CaMnSb_2$, $EuMnSb_2$, and $YbMnSb_2$ exhibit C-type AFM order, we expect that A^5MnSb_2 also has such C-type magnetic structure, which is used in our non-SOC calculations. However, the direction of magnetic moments differs among these four compounds: canted C-type AFM for $SrMnSb_2$, C-type AFM with moments along the c axis for $EuMnSb_2$, and C-type AFM with moments along the in-plane direction for $YbMnSb_2$. Consequently, another C-type ordering may be realized in the high-entropy crystal due to the compromise mechanism. Neutron diffraction studies for identifying the magnetic configuration and subsequent detailed calculation including SOC is a possible direction of future investigations.

Technical comment 17

Regarding DFT: Could the author explain in more detail why “the band dispersion measured along [Fig. 4(b)] corroborates the presence of the Dirac cone”? Even though all DFT calculations presented in this paper show step and linearly dispersing bands near the X point but none of these calculations actually show a band crossing (not even above E_F) that would qualify those linear dispersing bands as Dirac bands.

Response: We are thankful for the comment. The band dispersion of $AMnX_2$ family ($A = Ba, Sr, Ca, Eu, \text{ and } Yb; X = Sb \text{ and } Bi$) is dictated by the crystal structure (i.e., the chemistry of the A

cation) and the magnitude of spin-orbital coupling (i.e., the chemistry of the X anion), which was summarized in Klemenz *et. al.*, *Annual Review of Materials Research*, **49**, 185 (2019) (Ref. [28] of the main manuscript). The structures without X -displacement ($P4/nmm$ and $I4/mmm$ structures, see Fig. 1 of the main manuscript) exhibit Dirac crossing at the X point with SOC being excluded. Activating SOC leads to a small gap at the Dirac point; therefore the materials are no longer ideal Dirac semimetals but Dirac materials. The first established example of such a Dirac material is SrMnBi_2 [*Phys. Rev. Lett.* **107**, 126402] with a tetragonal structure ($I4/mmm$), for which massive Dirac fermion ($m^* = 0.29m_0$) was also probed by quantum oscillations. The X -displacement (which leads to zig-zag chain formation in X_4 layers) also results in the gap opening at the X point even without SOC. As observed in our STEM measurements, the high-entropy crystal involves Sb displacement in the Sb_4 layers, and such displacement was taken into account in the structural model used for the band calculations. This is why the Dirac point is gapped, and crossings cannot be found in the calculated band structure shown in Fig. 4(d). Due to the hole-doped nature, the band top of the gapped Dirac state was inaccessible by ARPES measurements. However, the steep linear $E(k)$ dispersion identified in the ARPES data well captured the feature of gapped Dirac states and was reasonably reproduced by the DFT-calculated band around the X point.

Technical comment 18

Regarding electrical transport: Since the author discuss the Hall data for almost one full paragraph it might make sense to show them in the main text.

Response: According to this comment, we have added the Hall resistivity data (ρ_{xy}) to the panel (a) of Figure 5 in the main text.

Technical comment 19

Regarding electrical transport: Can the authors please discuss the Zeeman splitting of the quantum oscillations in more detail? The discussion does not lead towards that feature and it appears quite unexpected in the text.

Response: We thank the reviewer for this valuable comment. Accordingly, we have added the following discussion to the Supplementary Information and indicated in the main text that the detailed discussion of the Zeeman splitting is contained in the Supplemental Information.

As mentioned in the main text, the Zeeman splitting of the quantum oscillation peaks is prominent at low temperatures (lower than 30 K). For clarity, we plot the first derivative of ρ_{xx} as a function of $1/B$ in Figure S3. For a particular Landau level, the spin-up and spin-down sub-Landau levels cross the Fermi energy at the magnetic field of B^- and B^+ , respectively. Using the following equation,

$$F_\alpha \left(\frac{1}{B^+} - \frac{1}{B^-} \right) = \frac{gm^*}{2m^0},$$

where F_α , m^* , m^0 , g is oscillation frequency, effective mass, static electron mass, and Landè g -factor, respectively, we obtained 1.55 for g . This value is small compared to other Dirac semimetals

such as ZrTe_5 and Cd_3As_2 , but similar to the estimated values for pristine BaMnSb_2 (≈ 2).

Technical comment 20

Regarding electrical transport: The use of the intercept of the Landau level fan diagram as a proof for 3D rather than 2D nature of the Fermi surface is not very convincing. Are there other possible explanations for the deviation of the intercept from 0.5?

Response: We think that the difference in chemical potential is another possible explanation for the deviation of the intercept from 0.5. As studied previously for BaMnSb_2 and its doped compositions [$\text{Ba}_{0.9}\text{Eu}_{0.1}\text{MnSb}_2$ (heavily hole-doped) and $\text{BaMn}_{0.9}\text{Zn}_{0.1}\text{Sb}_2$ (the least hole-doped), *Nature Communications* **12**,4062 (2021)], the change in chemical potential resulted in the change of Berry phase of about 0.1π , corresponding to the intercept difference of about 0.06. We understand that the intercepts obtained for the high-entropy crystal, 0.75(7) and 0.62(9) [Fig. R3], which are very close to 0.5 within the error, is not very convincing to use as proof for the 3D Fermi surface. However, please note that our primary evidence for the highly-anisotropic (or quasi-3D) Fermi surface is the angular dependence of the oscillation frequency, which deviates from the $1/\cos\theta$ curve unlike the case of 2D Fermi surfaces and that the deviation of the intercept just shows the consistency. We have revised the manuscript to clarify this point.

Reviewers' Comments:

Reviewer #1:

Remarks to the Author:

The authors have satisfactorily addressed my concerns and made corresponding modifications. The manuscript has been improved. I therefore recommend the publication.

Reviewer #2:

Remarks to the Author:

The authors have done a great job in addressing all comments. I therefore recommend publishing this article.

Reviewer #3:

Remarks to the Author:

We thank the authors for their thorough responses to our initial report. Our earlier concerns have all been fully addressed. In particular, the concern related to the identification of the non-centrosymmetric space group is unambiguously resolved by the inclusion of reference SHG data and complete single crystal refinement data.

This work is a very original and high-quality contribution to the field of high entropy intermetallics. It provides a template for the discovery of materials with novel structures. We, therefore, fully endorse the publication of this work in Nature Communications.

Reviewer #4:

Remarks to the Author:

► Reply to the comment of Reviewer #1

General comments

The authors have satisfactorily addressed my concerns and made corresponding modifications. The manuscript has been improved. I therefore recommend the publication.

Response: We would like to thank the reviewer for the positive comments. We are delighted to hear that our revised manuscript has been improved and thus is now recommended for publication in Nature Communications.

► Reply to the comments of Reviewer #2

General comments

The authors have done a great job in addressing all comments. I therefore recommend publishing this article.

Response: We would like to thank the reviewer for kindly reviewing our manuscript. We are glad to hear that we can share our results through the publication in Nature Communications.

► Reply to the comments of Reviewer #3

General comments

We thank the authors for their thorough responses to our initial report. Our earlier concerns have all been fully addressed. In particular, the concern related to the identification of the non-centrosymmetric space group is unambiguously resolved by the inclusion of reference SHG data and complete single crystal refinement data.

This work is a very original and high-quality contribution to the field of high entropy intermetallics. It provides a template for the discovery of materials with novel structures. We, therefore, fully endorse the publication of this work in Nature Communications.

Response: We would like to thank the reviewer for taking the time to review our manuscript and are delighted to hear that our response has fully addressed their concerns. We appreciate their positive judgment and endorsement for the publication in Nature Communications.

► Reply to the comments of Reviewer #4**General comments**

Response: We would like to thank the reviewer for helping one of the reviewers above. We are glad to hear that our manuscript provided an opportunity for the professional development of an early career researcher in addition to the scientific contribution to the community.